# Towards Efficient Training of Graph Neural Networks: A Multiscale Approach

**Eshed Gal**                                                    *eshedg@post.bgu.ac.il*
*Faculty of Computer and Information Science*
*Ben-Gurion University of the Negev*

**Moshe Eliasof**                                                    *me532@cam.ac.uk*
*Department of Applied Mathematics and Theoretical Physics*
*University of Cambridge*

**Carola-Bibiane Schönlieb**                                          *cbs31@cam.ac.uk*
*Department of Applied Mathematics and Theoretical Physics*
*University of Cambridge*

**Ivan I. Kyrchei**                                              *ivankyrchei26@gmailcom*
*Pidstryhach Institute for Applied Problems of Mechanics and Mathematics*
*NAS of Ukraine, L'viv, Ukraine*

**Eldad Haber**                                                   *ehaber@eoas.ubc.ca*
*Department of Earth, Ocean and Atmospheric Sciences*
*University of British Columbia*

**Eran Treister**                                                    *erant@bgu.ac.il*
*Faculty of Computer and Information Science*
*Ben-Gurion University of the Negev*

**Reviewed on OpenReview:** *https://openreview.net/forum?id=2eZ8xkL2ZB*

## Abstract

Graph Neural Networks (GNNs) have become powerful tools for learning from graph-structured data, finding applications across diverse domains. However, as graph sizes and connectivity increase, standard GNN training methods face significant computational and memory challenges, limiting their scalability and efficiency. In this paper, we present a novel framework for efficient multiscale training of GNNs. Our approach leverages hierarchical graph representations and subgraphs, enabling the integration of information across multiple scales and resolutions. By utilizing coarser graph abstractions and subgraphs, each with fewer nodes and edges, we significantly reduce computational overhead during training. Building on this framework, we propose a suite of scalable training strategies, including coarse-to-fine learning, subgraph-to-full-graph transfer, and multiscale gradient computation. We also provide some theoretical analysis of our methods and demonstrate their effectiveness across various datasets and learning tasks. Our results show that multiscale training can substantially accelerate GNN training for large-scale problems while maintaining, or even improving, predictive performance.

## 1 Introduction

Large-scale graph networks arise in diverse fields such as social network analysis (Kipf & Welling, 2016; Defferrard et al., 2016), recommendation systems (Tang et al., 2018), and bioinformatics (Jumper et al., 2021), where relationships between entities are crucial to understanding system behavior (Abadal et al., 2021). These

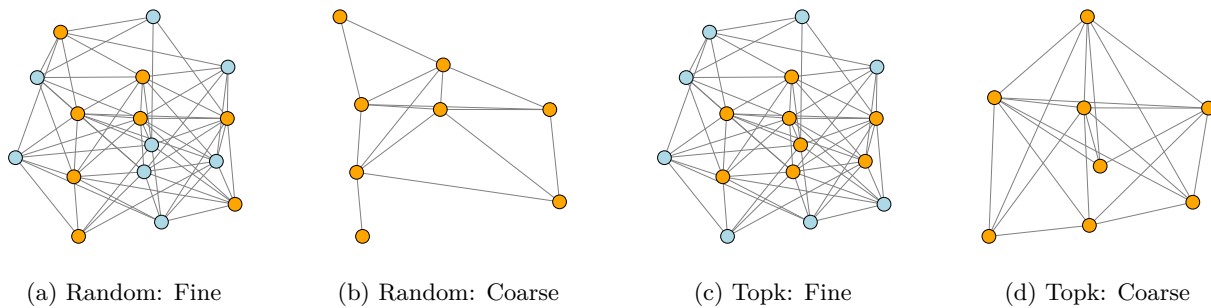

(a) Random: Fine      (b) Random: Coarse      (c) Topk: Fine      (d) Topk: Coarse

Figure 1: Graph coarsening examples. Left two: **random** pooling; Right two: **Topk** pooling (highest-degree nodes selected). Orange nodes indicate selected coarse nodes.

problems involve processing large graph datasets, necessitating scalable solutions. Some examples of those problems include identifying community structures in social networks (Hamilton et al., 2017), optimizing traffic flow in smart cities (Geng et al., 2019), or studying protein-protein interactions in biology (Xu et al., 2024). In all these problems, efficient processing of graph data is required for practical reasons.

In recent years, many Graph Neural Networks (GNNs) algorithms have been proposed to solve large graph problems, such as GraphSage (Hamilton et al., 2017) and FastGCN (Chen et al., 2018). However, despite their success, when the involved graph is of a large scale, with many nodes and edges, it takes a long time and many resources to train the network (Galmés et al., 2021).

In this paper, we study and develop multiscale training methodologies, to improve the training efficiency of GNNs. To achieve that, we focus on the standard approach for implementing GNNs, that is, the Message-Passing (MP) scheme (Gilmer et al., 2017). Mathematically, this approach is equivalent to the multiplication of the graph adjacency matrix $\mathbf{A}$ with node feature maps $\mathbf{X}^{(l)}$, which is then multiplied by a learned channel-mixing weight matrix $\mathbf{W}^{(l)}$, followed by an application of an element-wise nonlinear activation function $\sigma$, as follows:

$$\mathbf{X}^{(l+1)} = \sigma(\mathbf{A}\mathbf{X}^{(l)}\mathbf{W}^{(l)}). \tag{1}$$

Assuming that the adjacency matrix is sparse with an average sparsity of $k$ neighbors per node, and a channel-mixing matrix of shape $c \times c$, we now focus on the total multiplications in the computation of Equation (1), to which we refer to as *'work'*. In this case, the work required in a single MP layer is of order $\mathcal{O}(n \cdot k \cdot c^2)$, where $n$ is the number of nodes in the graph, and $c$ is the number of channels. For large or dense graphs, i.e., $n$ or $k$ are large, the MP in Equation (1) requires a high computational cost. Any algorithm that works with the entire graph at once is therefore bound to face the cost of this matrix-vector product.

A possible solution to this high computational cost involves the utilization of a *multiscale framework*, which requires the generation of surrogate problems to be solved (Kolaczyk & Nowak, 2003). If the solution of the surrogate problem approximates the solution of the fine (original) scale problem, then it can be used as an initial starting point, thus reducing the work required to solve the optimization problem. In the context of graphs, such a multiscale representation of the original problem can be achieved using *graph coarsening*. Graph coarsening algorithms are techniques that reduce the size of graphs while preserving their essential properties (Cai et al., 2021). These methods aim to create smaller, more computationally manageable graphs by choosing a subset of the nodes or the edges, based on certain policies,

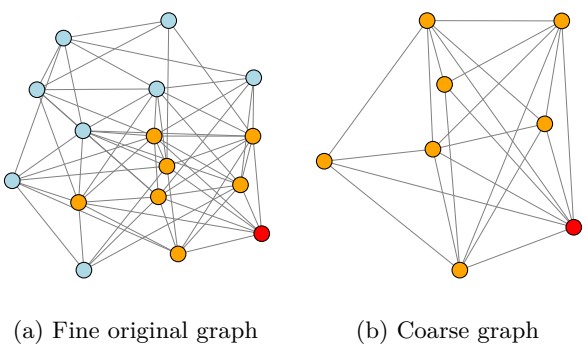

(a) Fine original graph      (b) Coarse graph

Figure 2: Coarse graph using **subgraph** pooling. Orange nodes indicate selected coarse nodes; the red node is the root for the ego-network.

or by using a subgraph of the original graph, and
may be implemented in various ways.

We demonstrate the generation of a coarse graph using random pooling and Topk pooling (which is based on the nodes with high degrees in the graph) in Figure 1. As for choosing a subgraph, this can be implemented using node-based selection methods, such as ego-networks (Gupta et al., 2014) or node marking and deletion (Frasca et al., 2022), which offer straightforward ways to extract subgraphs based on specific nodes of interest. We show an example of subgraph coarsening in Figure 2.

**Contribution.** In this paper, we propose *novel graph multiscale optimization algorithms* to obtain computationally efficient GNN training methods that retain the performance of standard gradient-based training of GNNs. Our approach is based on a multiscale framework that utilizes various graph coarsening techniques. While coarsening and subgraph methods have been in use for various purposes, from improved architectures (Gao & Ji, 2019), to enhanced expressiveness (Bevilacqua et al., 2022; 2024), to the best of our knowledge, they were not considered in the context of efficient training of GNNs. Therefore, in this work, we explore three novel training mechanisms to design surrogate processes that approximate the original training problem with a fraction of the computational cost, using a multiscale approach. All the methods attempt to use the full graph with $n$ nodes as little as possible, and focus on training the network using coarse-scale graphs, thus reducing computational cost. The three methods, detailed in Section 4, are as follows:

- **Coarse-to-Fine.** A coarse-to-fine algorithm approximates the optimization problem on smaller, pooled graphs. If the coarse-scale solution is similar to the fine-scale one, it can serve as an effective initialization for the fine-scale optimization process. Since gradient-based methods converge faster when initialized near the solution (Arora et al., 2018; Boyd & Vandenberghe, 2004), this approach reduces the overall number of iterations required.
- **Sub-to-Full.** This approach leverages a sequence of subgraphs that progressively grow to the original fine-scale graph. Like coarsening, each subgraph is significantly smaller than the original graph, enabling faster optimization process. Subgraphs can be generated based on node distances (if available) or ego-network methods (Gupta et al., 2014).
- **Multiscale Gradients Computation.** Building on the previous methods and inspired by Multiscale Monte Carlo (Giles, 2001), which is widely used for solving stochastic differential equations, we introduce a mechanism to approximate fine-scale gradients at a fraction of the computational cost, achieved by combining gradients across scales to efficiently approximate the fine-scale gradient.

To demonstrate their effectiveness, we benchmark the obtained performance and reduce computational costs on multiple graph learning tasks, from transductive node classification to inductive tasks. Our code is available here: https://github.com/eshedgal1/GraphMultiscale.

## 2 Related Work

We discuss related topics to our paper, from hierarchical graph pooling to multiscale methods.

**Graph Pooling.** Graph pooling can be performed in various ways, as reviewed in Liu et al. (2022). Popular pooling techniques include DiffPool (Ying et al., 2018) MaxCutPool (Abate & Bianchi, 2024), graph wavelet compression (Eliasof et al., 2023), as well as Independent-Set Pooling (Stanovic et al., 2024), all aiming to produce a coarsened graph that bears similar properties to the original graph. Wang et al. (2020) and Wang et al. (2024) introduce the concept of subgraph pooling to GNNs. Bianchi & Lachi (2023) study the expressive power of pooling in GNNs. Here, we use graph pooling mechanisms, as illustrated in Figure 1–Figure 2, as core components for efficient training.

**General Multiscale Methods.** The idea of Coarse-to-Fine for the solution of optimization problems has been proposed in Brandt (1984), while a more detailed analysis of the idea in the context of optimization problems was introduced in Nash (2000). Multiscale approaches are also useful in the context of Partial

Differential Equations (Guo & Li, 2024; Li et al., 2024b), as well as for various applications, including flood modeling (Bentivoglio et al., 2024). Lastly, Monte-Carlo methods (Hastings, 1970) have also been adapted to a multiscale framework (Giles, 2001), which can be used for various applications such as electrolytes and medical simulations (Liang et al., 2015; Klapproth et al., 2021). In this paper, we draw inspiration from multiscale methods to introduce a novel and efficient GNN training framework.

**Multiscale Learning.** In the context of graphs, the recent work in Shi et al. (2022) has shown that a matching problem can be solved using a coarse-to-fine strategy, and Cai et al. (2021) proposed different coarsening strategies. Other innovations include the recent work of Li et al. (2024a), whose goal is to mitigate the lack of labels in semi-supervised and unsupervised settings. We note that it is important to distinguish between *algorithms that use a multiscale structure as a part of the network*, such as Unet (Ronneberger et al., 2015) and Graph-Unet (Gao & Ji, 2019), and *multiscale training that utilizes multiscale for training*, such as the recent work on convolutional neural networks in Ahamed et al. (2025). In particular, the latter, multiscale training, which is the focus of this paper, can be potentially applied to any GNN architecture.

## 3 Preliminaries and Notations

We now provide related graph-learning notations and preliminaries that we will use throughout this paper. Let us consider a graph defined by $\mathcal{G} = (\mathcal{V}, \mathcal{E})$ where $\mathcal{V}$ is a set of $n$ nodes and $\mathcal{E}$ is a set of $m$ edges. Alternatively, a graph can be represented by its adjacency matrix $\mathbf{A}$, where the $(i, j)$-th entry is 1 if an edge between nodes $i$ and $j$ exists, and 0 otherwise. We assume that each node $i$ is associated with a feature vector with $c$ channels, $\mathbf{x}_i \in \mathbb{R}^c$, and $\mathbf{X} = [\mathbf{x}_0 \ldots, \mathbf{x}_{n-1}] \in \mathbb{R}^{n \times c}$ is the node features matrix. We denote by $\mathbf{y}_i \in \mathbb{R}^d$ the $i$-th node label, and by $\mathbf{Y} = [\mathbf{y}_0, \ldots, \mathbf{y}_{n-1}] \in \mathbb{R}^{n \times d}$ the label matrix.

In a typical graph learning problem, we are given graph data $(\mathcal{G}, \mathbf{X}, \mathbf{Y})$, and our goal is to learn a function $f(\mathbf{X}, \mathcal{G}; \boldsymbol{\theta})$, with learnable parameters $\boldsymbol{\theta}$ such that

$$\mathbf{Y} \approx f(\mathbf{X}, \mathcal{G}; \boldsymbol{\theta}). \tag{2}$$

For instance, in the well-known Graph Convolution Network (GCN) (Kipf & Welling, 2016), each layer is defined using Equation (1), such that $\mathbf{X}^{(l+1)}$ is the output of the $l$-th layer and serves as the input for the $(l + 1)$-th layer. For a network with $L$ layers, we denote by $\boldsymbol{\theta} = \{\mathbf{W}^{(1)}, \ldots, \mathbf{W}^{(L)}\}$, the collection of all learnable weight matrices within the network.

It is important to note that the matrix $\mathbf{W}^{(l)}$ is independent of the size of the adjacency and node feature matrices, and its dimensions only correspond to the feature space dimensions of the input and output layer. In terms of computational costs, a the $l$-th GCN layer requires the multiplication of the three matrices $\mathbf{A}, \mathbf{X}^{(l)}$ and $\mathbf{W}^{(l)}$.

We note that the number of multiplications computed in the $l$-th GCN layer is directly affected by the size of the node features matrix $\mathbf{X}^{(l)}$, and the graph adjacency matrix $\mathbf{A}$. This means that for a very large graph $\mathcal{G}$, each layer requires significant computational resources, both in terms of time and memory usage. These computational costs come into effect both during the training process and at the inference stage. In this paper, we directly address these associated costs by leveraging GNN layers with smaller matrices $\mathbf{A}_r$ and $\mathbf{X}_r^{(l)}$, yielding trained networks whose downstream performance is similar to those that would have been achieved had we used the original input graph and features, while significantly reducing the computational costs.

Lastly, we note that, as shown in Section 6, our approach is general and applicable to various GNN architectures.

## 4 Efficient Training Framework for GNNs

**Overarching goal.** We consider the graph-learning problem in Equation (2), defined on a graph with $n$ nodes. To reduce the cost of the learning process, we envision a sequence of $R$ surrogate problems, defined by the set of data tuples $\{(\mathcal{G}_r, \mathbf{X}_r, \mathbf{Y}_r)\}_{r=1}^R$ and problems of the form

$$\mathbf{Y}_r \approx f(\mathbf{X}_r, \mathcal{G}_r; \boldsymbol{\theta}), \quad r = 1, \ldots, R \tag{3}$$

---

**Algorithm 1** Multiscale algorithm

---

   *# R – total number of levels.*
   Initialize weights $\boldsymbol{\theta}$ and network $f$.
   **for** $r = R, R - 1, \ldots, 1$ **do**
      Compute the coarse graph $\mathcal{G}_r$ that has $n_r$ nodes.
      Compute the coarse node features & labels $(\mathbf{X}_r, \mathbf{Y}_r)$.
      Solve the optimization problem in Equation (4).
      Update the weights $\boldsymbol{\theta}$.
   **end for**

---

where $f$ is a GNN and $\mathcal{G}_r$ is a graph with $n_r$ nodes, where $n = n_1 > n_2 > \ldots > n_R$. Each graph $\mathcal{G}_r$ is obtained by pooling the fine graph $\mathcal{G}$, using a smaller set of nodes compared to the previous (finer) graph $\mathcal{G}_{r-1}$. $\mathbf{X}_r$ is the node features matrix of the graph $\mathcal{G}_r$, and $\mathbf{Y}_r$ is the corresponding label matrix. In what follows, we explore different ways to obtain $\mathcal{G}_r, \mathbf{X}_r$ and $\mathbf{Y}_r$ from the original resolution variables.

An important feature of GNNs is that the weights $\boldsymbol{\theta}$ of the network are independent of the input and output size. Each weight matrix $\mathbf{W}^{(l)}$ is of dimensions $c_{in} \times c_{out}$, for a $\mathbf{X}^{(l)} \in \mathbb{R}^{n \times c_{in}}$ node feature matrix of the layer $l$, and $\mathbf{X}^{(l+1)} \in \mathbb{R}^{n \times c_{out}}$ for the corresponding dimensions of the next layer $l + 1$. This means that changing the number of nodes and edges in the graph will not affect the weight matrix dimensions while reducing the number of computational operations conducted in the learning process. That is, the set of parameters $\boldsymbol{\theta}$ can be used on any of the $R$ problems in Equation (3). Let us reformulate the learning problem on the $r$-th level as follows:

$$\widehat{\boldsymbol{\theta}}_r = \text{argmin}_{\boldsymbol{\theta}} \, \mathbb{E}[\, \mathcal{L} \left( f(\mathbf{X}_r, \mathcal{G}_r; \boldsymbol{\theta}), \mathbf{Y}_r \right)], \tag{4}$$

where $\mathcal{L}$ is the loss function that measures the error in Equation (3), and the expectation is on the data $(\mathbf{X}_r, \mathbf{Y}_r)$. Our goal is to use the optimization problem in Equation (4) as a cheap surrogate problem to obtain an approximate solution (i.e., weights $\boldsymbol{\theta}$) of the original problem in Equation (2). We describe three methodologies to achieve this goal.

### 4.1 Coarse-to-Fine Training

One seemingly simple idea is to solve the $r$-th surrogate problem in Equation (4) and use the obtained weights to solve the problem on the $(r - 1)$-th graph, that is $\mathcal{G}_{r-1}$, which is larger. This approach can be beneficial only if $\widehat{\boldsymbol{\theta}}_{r+1} \approx \widehat{\boldsymbol{\theta}}_r$, that is, the obtained optimal weights are similar under different scales, and if the total amount of work on the coarse scale is smaller than on the fine one, such that it facilitates computational cost reduction. The multiscale framework is summarized in Algorithm 1.

While Algorithm 1 is relatively simple, it requires obtaining the $r$-th surrogate problem by coarsening the graph, as well as a stopping criterion for the $r$-th optimization problem. Generally, in our coarse-to-fine method, we use the following graph coarsening scheme:

$$\mathbf{A}_r = \mathbf{P}_r^T \mathbf{A}^p \mathbf{P}_r, \tag{5}$$

where $\mathbf{P}_r \in \mathbb{R}^{n \times n_r}$ is a binary injection matrix that maps the fine graph with $n$ nodes to a smaller graph with $n_r$ nodes. That is, we choose a submatrix of the $p$-th power of the adjacency matrix $\mathbf{A}$. We use $\mathbf{X}_r = \mathbf{P}_r^T \mathbf{X}$ as the coarse graph node features matrix and $\mathbf{Y}_r = \mathbf{P}_r^T \mathbf{Y}$ as the label matrix of the coarse graph.

Choosing the non-zero entries of the matrix $\mathbf{P}_r$, which are the nodes that remain in the graph after pooling, can be implemented in various ways. For example, $\mathbf{P}_r$ can be obtained by sampling random nodes of the fine graph $\mathcal{G}$, as illustrated in Figure 1 (left), or with some desired pooling logic, such as choosing nodes of the highest degree in the graph, which is a method often described as $Topk$ pooling (Huang et al., 2015) and we present an illustration of it in Figure 1 (right). The power $p$ in Equation (5) controls the connectivity of the graph, where $p = 1$ is the original graph adjacency matrix, and using a value $p > 1$ will enhance the connectivity of the graph prior to the coarsening step. We note that the range of applicable choices for $p$ is subject to the given data, as $\mathbf{A}_r$ has to contain fewer edges than the original matrix $\mathbf{A}$ in order for the resulting graph to be coarse.

In Algorithm 1, we apply GNN layers such as in Equation (1), using the coarsened feature matrix $\mathbf{X}_r$ and its corresponding adjacency matrix $\mathbf{A}_r$, that is based on a desired pooling ratio. We note that there is no re-initialization of the weight matrix $\mathbf{W}$ when transferring from coarse grids to finer ones, allowing a continuous learning process along the steps of the algorithm. As we show later in our experiments in Section 6, our coarse-to-fine training approach, summarized in Algorithm 1, is not restricted to a specific GNN architecture or a pooling technique.

### 4.2 Sub-to-Full Training

We now describe an additional method for efficient training on GNNs, that relies on utilizing subgraphs, called Sub-to-Full. At its core, the method is similar to Coarse-to-Fine, as summarized in Algorithm 1, using the same coarsening in Equation (5). However, its main difference is that we now choose the indices of $\mathbf{P}_r$ such that the nodes $\mathbf{X}_r$ form a subgraph of the original graph $\mathcal{G}$.

A subgraph can be defined in a few possible manners. One is by choosing a root node $j$, and generating a subgraph centered around that node. If the graph $\mathcal{G}$ has some geometric features, meaning each input node corresponds to a point in the Euclidean space $\mathbb{R}^d$, one might consider constructing a subgraph with $n_r$ nodes to be the $n_r$ nodes that are closest to the chosen center node in terms of Euclidean distance norm. Otherwise, we can consider a subgraph created using an ego-network (Gupta et al., 2014) of $k - hop$ steps, such that we consider all neighbors that are $k$ steps away from the central node. An illustration of this method is presented in Figure 2. The number of hops, $k$, is a hyperparameter of the algorithm process, and we increase it as we go from coarse levels to fine ones within the series of graph $\{\mathcal{G}_r\}_{r=1}^R$.

### 4.3 Multiscale Gradients Computation

In Section 4.1–Section 4.2, we presented two possible methods to facilitate the efficient training of GNNs via a multiscale approach. In what follows, we present a complementary method that builds on these ideas, as well as draws inspiration from Monte-Carlo optimization techniques (Giles, 2001), to further reduce the cost of GNN training.

We start with the observation that optimization in the context of deep learning is dominated by stochastic gradient-based methods (Kingma & Ba, 2014; Ruder, 2016). At the base of such methods, stands the gradient descent method, where at optimization step $s$ we have:

$$\boldsymbol{\theta}_{s+1} = \boldsymbol{\theta}_s - \mu_s \mathbb{E}(\nabla \mathcal{L}\left(f(\mathbf{X}_1, \mathcal{G}_1; \boldsymbol{\theta}_s)\right)). \tag{6}$$

However, when large datasets are considered, it becomes computationally infeasible to take the expectation of the gradient with respect to the complete dataset. This leads to the *stochastic* gradient descent (SGD) method, where the gradient is approximated using a smaller batch of examples of size $B$, obtaining the following weight update equation:

$$\boldsymbol{\theta}_{s+1} = \boldsymbol{\theta}_s - \frac{\mu_s}{B} \sum_{b=1}^{B} \nabla \mathcal{L}\left(f(\mathbf{X}_1^{(b)}, \mathcal{G}_1^{(b)}; \boldsymbol{\theta})\right), \tag{7}$$

where $\mathbf{X}_1^{(b)}, \mathcal{G}_1^{(b)}$ denote the node features and graph structure of the $b$-th element in the batch.

To compute the update in Equation (7), one is required to evaluate the loss $\mathcal{L}$ for $B$ times on the finest graph $\mathcal{G}_1 = \mathcal{G}$, which can be computationally demanding. To reduce this computational cost, we consider the following telescopic sum identity:

$$\mathbb{E}\,\mathcal{L}_s^{(1)}(\boldsymbol{\theta}) = \mathbb{E}\,\mathcal{L}_s^{(R)}(\boldsymbol{\theta}) + \mathbb{E}\left(\mathcal{L}_s^{(R-1)}(\boldsymbol{\theta}) - \mathcal{L}_s^{(R)}(\boldsymbol{\theta})\right) + ... + \mathbb{E}\left(\mathcal{L}_s^{(1)}(\boldsymbol{\theta}) - \mathcal{L}_s^{(2)}(\boldsymbol{\theta})\right), \tag{8}$$

where $\mathcal{L}_s^{(r)} = \mathcal{L}\left(f(\mathbf{X}_r, \mathcal{G}_r; \boldsymbol{\theta}), \mathbf{Y}_r\right)$ represents the loss of the $r$-th scale at the $s$-th optimization step. Equation (8) induces a *hierarchical stochastic gradient descent* computation based on the set of multiscale graph $\{\mathcal{G}_r\}_{r=1}^R$.

---

**Algorithm 2** Multiscale Gradients Computation

---

$\quad$ # R – total number of levels.
$\quad$ Initialize $\mathcal{L} = 0$
$\quad$ **for** $r = 2, \ldots, R$ **do**
$\quad\quad$ Sample $M_r$ node subsets
$\quad\quad$ Compute $\mathbf{X}_r, \mathbf{X}_{r-1}, \mathbf{Y}_r, \mathbf{Y}_{r-1}$
$\quad\quad$ Compute the graphs $\mathcal{G}_r$ and $\mathcal{G}_{r-1}$
$\quad\quad$ Update $\mathcal{L} \leftarrow \mathcal{L} + \frac{1}{M_r}(\mathcal{L}^{(r-1)} - \mathcal{L}^{(r)})$
$\quad$ **end for**
$\quad$ Compute $\mathcal{L}^{(R)}$ and update $\mathcal{L} \leftarrow \mathcal{L} + \frac{1}{M_R}\mathcal{L}^{(R)}$

---

The main idea of the hierarchical stochastic gradient descent method in Equation (8) is that each expectation in the telescopic sum can use a different number of realizations for its approximation. The assumption is that we can choose hierarchies such that

$$\left| \mathcal{L}_s^{(r-1)}(\boldsymbol{\theta}) - \mathcal{L}_s^{(r)}(\boldsymbol{\theta}) \right| \leq \gamma_r \cdot \mathcal{L}_s^{(r-1)}(\boldsymbol{\theta}) \tag{9}$$

for some $1 > \gamma_r > \gamma_{r-1} > \ldots > \gamma_1 = 0$. To understand why this concept is beneficial, let us consider the discretization of the fine-scale problem (i.e., for $\mathcal{G}_1$) in Equation (4) using $M$ samples. Suppose that the error in approximating the expected loss at the first, finest scale $\mathbb{E}[\mathcal{L}_s^{(1)}]$ by the samples behaves as $e^{(1)} = \frac{C}{\sqrt{M}}$. Now, let us consider a 2-scale process, with $M_1$ samples on the finest resolution graph. The error in evaluating the loss on the second scale $\mathbb{E}[\mathcal{L}_s^{(2)}]$ with $M_2$ examples is given by $e^{(2)} = \frac{C}{\sqrt{M_2}}$ while the error in evaluating $\mathbb{E}[(\mathcal{L}_s^{(1)} - \mathcal{L}_s^{(2)})]$ is $e^{(1,2)} = \frac{C\gamma_1}{\sqrt{M_1}}$. Since the assumption is that $\gamma < 1$, we are able to choose $M_2$ and $M_1$ such that $e^{(1)} \approx e^{(2)} + e^{(1,2)}$. Hence, rather than using $M$ finest graph resolution observations, we use only $M_1 < M$ such observations and sample the coarser problems instead.

More broadly, we can generalize the case of a 2-scale optimization process to an $R$ multiscale process, such that an approximation of the fine-scale loss $\mathcal{L}_s^{(1)}$ is obtained by:

$$\mathbb{E}\,\mathcal{L}_s^{(1)}(\boldsymbol{\theta}) \quad \approx \quad \frac{1}{M_R}\sum \mathcal{L}_j^{(R)}(\boldsymbol{\theta}) + \frac{1}{M_{R-1}}\sum \left( \mathcal{L}_s^{(R-1)}(\boldsymbol{\theta}) - \mathcal{L}_s^{(R)}(\boldsymbol{\theta}) \right) + \frac{1}{M_1}\sum \left( \mathcal{L}_s^{(1)}(\boldsymbol{\theta}) - \mathcal{L}_s^{(2)}(\boldsymbol{\theta}) \right), \tag{10}$$

where $M_R > M_{R-1} > M_1$. This implies that for the computation of the loss on the fine resolution $\mathcal{G}_1 = \mathcal{G}$, only a few fine graph approximations are needed, and most of the computations are done on coarse, small graphs. Our method is summarized in Algorithm 2, and we provide an illustration of the computation process in Figure 3. Naturally, this method suits inductive learning tasks, where multiple graphs are available.

## 4.4 A Motivating Example

For multiscale training methods to adhere to the original learning problem, the condition in Equation (9) needs to be satisfied. This condition is dependent on the chosen pooling strategy. To study the impact of different pooling strategies, we explore a simplified model problem. We construct a dataset, illustrated in Figure 4, that consists of structured synthetic graphs representing rods with three distinct types: (1) rods with both endpoints colored blue, (2) rods with both endpoints colored yellow, and (3) rods with mismatched endpoint colors. We wish to classify each node as either part of a rod or background and to identify the rod type.

A key challenge in this task is ensuring that the model can effectively capture global structure. To accurately classify a rod, a graph neural network (GNN) must aggregate information from both ends of the structure. A shallow GNN, such as one with a single or two-layer message-passing scheme, is insufficient, as its receptive field does not extend across the entire rod. Additionally, an effective coarsening strategy must preserve key structural information—retaining nodes from both rod endpoints, central regions, and the background—to maintain an informative representation.

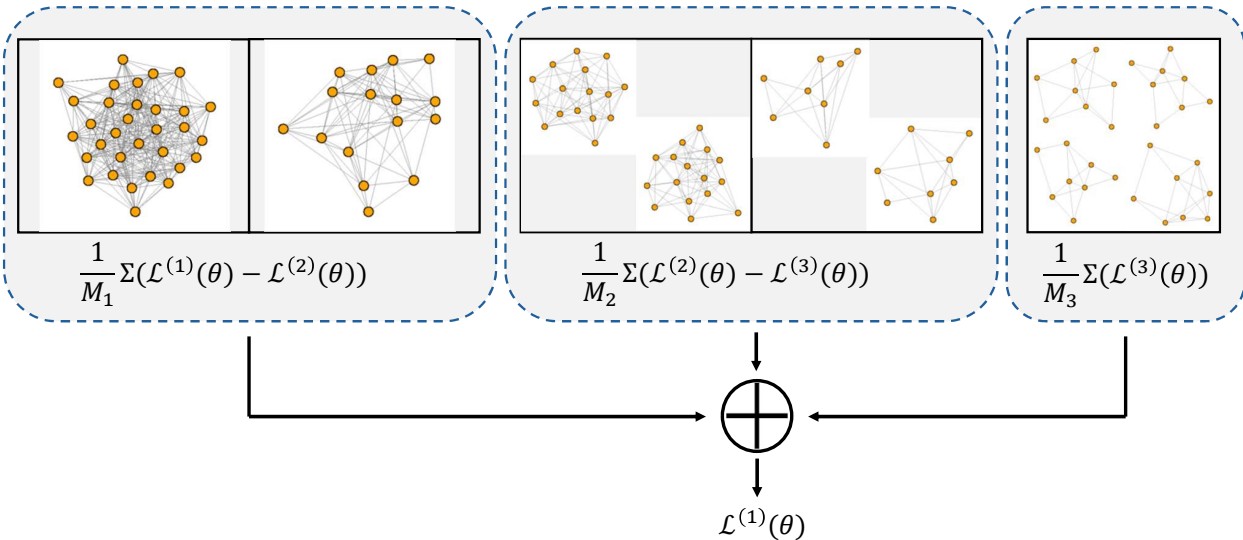

$$\frac{1}{M_1}\Sigma(\mathcal{L}^{(1)}(\theta) - \mathcal{L}^{(2)}(\theta)) \qquad \frac{1}{M_2}\Sigma(\mathcal{L}^{(2)}(\theta) - \mathcal{L}^{(3)}(\theta)) \qquad \frac{1}{M_3}\Sigma(\mathcal{L}^{(3)}(\theta))$$

$$\mathcal{L}^{(1)}(\theta)$$

Figure 3: Illustration of the Multiscale Gradient Computation algorithm introduced in Section 4.3.

Table 1: Experiments for different coarsening strategies, creating coarse graph with $\frac{1}{4}$ of the original nodes. Loss discrepancy between the loss computed on the original graph, and the pooled graph ($\Delta$loss) is calculated using $\gamma_s$ in Equation (9), and $p$ denotes the power in Equation (5).

| Coarsening type | # edges | $\Delta$loss (initial) | $\Delta$loss (final) |
|---|---|---|---|
| No coarsening | 41,358 | 0 | 0 |
| Random(p=1) | 3,987 | 7.72e-01 | 3.45e-01 |
| Random(p=2) | 6,673 | 5.62e-02 | 1.25e-02 |
| Topk(p=1) | 8,731 | 1.41e-01 | 2.53e-02 |
| Topk(p=2) | 20,432 | 3.12e-03 | 8.64e-03 |
| Subgraph | 9,494 | 1.25e-01 | 1.93e-01 |

To explore this further, we train a four-layer GCN (Kipf & Welling, 2016) on a dataset of 100 rods in varying orientations and evaluate different coarsening methods for constructing a reduced graph. Table 1 presents the loss values for various coarsening strategies along with the corresponding edge counts. The results highlight another aspect of our approach: methods that do not increase connectivity prior to pooling (i.e., with $p = 1$) may, in certain situations, lead to significant variance in loss across different coarsened graphs. For this dataset, introducing additional connectivity before pooling successfully mitigates this issue while still reducing the graph size in terms of both nodes and edges.

To further evaluate the capabilities of our multiscale methods on this synthetic dataset, Appendix B presents the full multiscale training results and compares them with standard fine-grid training. Additionally, we construct a second synthetic dataset derived from the MNIST image dataset (LeCun et al., 1998). Results on both datasets demonstrate that our methods effectively capture global graph features while enabling efficient training.

### 4.5 Theoretical Properties of Reduced Graphs

At the base of all the above-discussed techniques, the core is the ability of the coarse graph to represent the fine graph. To this end, consider the least squares model problem, in which we aim to minimize over $\boldsymbol{\theta}$ the

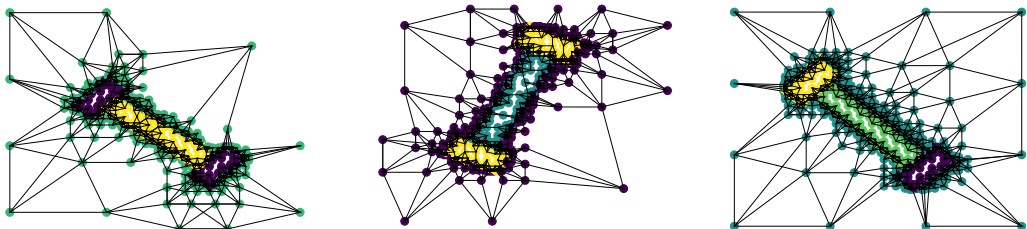

Figure 4: The Q-tips data set. The data contains three types of a "q-tip" and the goal is to label the stick as type 1,2 or 3.

following loss function:

$$\mathcal{L}(\boldsymbol{\theta}) = \frac{1}{2N}\|\sigma(\mathbf{A}\mathbf{X}\{\boldsymbol{\theta}\} - \mathbf{Y}\|^2, \tag{11}$$

and assume that $\sigma$ is the identity mapping. One way to analyze the solution of the coarse-scale problem is through sketching (see Woodruff (2014)). In sketching, the original problem is replaced by

$$\mathcal{L}_c(\boldsymbol{\theta}) = \frac{1}{2N}\|\mathbf{P}^\top\mathbf{A}\mathbf{X}\boldsymbol{\theta} - \mathbf{P}^\top\mathbf{Y}\|^2 \tag{12}$$

where $\mathbf{P}^\top$ is a sketching matrix. We establish the following result with its proof provided in Appendix A, which applies to Sub-to-Full training, and we note is only limited to the linear case $\sigma = Id$:

**Theorem 4.1.** *Let $\boldsymbol{\theta}^*$ be the solution of equation (11) and let $\boldsymbol{\theta}_C^*$ be the solution of equation (12). Denote $\epsilon > 0$, and let $\mathbf{P}^\top$ be a sparse subspace embedding with $O(\frac{c^2}{\epsilon})$ rows, where c is the size of the feature vectors (number of graph channels), that injects any vector into a subset of indices, which we denote by C. According to this subset, denote $\mathbf{A} = \begin{bmatrix} \mathbf{A}_{CC} & \mathbf{A}_{CF} \\ \mathbf{A}_{FC} & \mathbf{A}_{FF} \end{bmatrix}$, and $\mathbf{X} = \begin{bmatrix} \mathbf{X}_C \\ \mathbf{X}_F \end{bmatrix}$. Then if $\sigma = Id$, there exists $\|\mathbf{R}\| < \|\mathbf{A}_{CF}\mathbf{X}_F\boldsymbol{\theta}_C^*\|$ such that with probability .99:*

$$\mathcal{L}_c(\boldsymbol{\theta}_C^*) = \frac{1}{2N}\|\mathbf{P}^\top\mathbf{A}\mathbf{P}\mathbf{P}^\top\mathbf{X}\boldsymbol{\theta}_C^* - \mathbf{P}^\top\mathbf{Y}\|^2 \leq \frac{1 \pm \epsilon}{2N}\left(\|\mathbf{A}\mathbf{X}\boldsymbol{\theta}^* - \mathbf{Y}\|^2 + \|\mathbf{R}\|^2\right) \tag{13}$$

## 5 Computational Complexity

As noted in Section 3, the computational cost of GNN message-passing layers is directly influenced by the size of both the node feature matrix and the adjacency matrix. This observation suggests that reducing the number of nodes and edges can lead to improved computational efficiency during training.

To illustrate this, consider Graph Convolutional Networks (GCNs) (Kipf & Welling, 2016), where each layer follows the update rule defined in Equation 1. When analyzing floating-point operations (FLOPs), we account for both the input and output dimensions of each layer, as well as the sparsity of the adjacency matrix, meaning that only nonzero entries contribute to computation. The total number of FLOPs required for a single GCN layer is given by:

$$2 \cdot |\mathcal{E}| \cdot c_{in} + |\mathcal{V}| \cdot c_{in} \cdot c_{out}, \tag{14}$$

where $\mathcal{E}$ represents the set of edges, $\mathcal{V}$ the set of nodes, $c_{in}$ the input feature dimension, and $c_{out}$ the output feature dimension of the layer. From this expression, it follows that reducing the number of nodes and edges leads to a decrease in computational cost. For instance, if the number of nodes is halved, i.e., $|\mathcal{V}_{\mathcal{C}}| = \frac{1}{2}|\mathcal{V}|$, and the number of edges satisfies $|\mathcal{E}_{\mathcal{C}}| < |\mathcal{E}|$ for any pooling strategy, then the FLOPs per layer are reduced accordingly. This reduction translates into a significant theoretical improvement in efficiency.

Importantly, this effect is not limited to GCNs. A similar computational advantage is observed for other GNN architectures, like Graph Isomorphism Networks (GIN) (Xu et al., 2018) and Graph Attention Networks (GAT) (Velickovic et al., 2017). In all cases, reducing the graph size results in fewer computations per layer, demonstrating a generalizable advantage across different GNN architectures.

# 6 Experiments

**Objectives.** We demonstrate that our approach achieves comparable or superior downstream performance to standard GNN training on the original graph while reducing computational costs, across the training techniques outlined in Section 4 and the pooling strategies discussed below. We demonstrate our methods on both mid-size and large-scale datasets, showing that given a need to solve a certain graph problem, our methods will prove to be efficient given existing computational resources.

**Pooling Strategies.** We incorporate various coarsening techniques to enhance GNN training efficiency:

(i) **Random Coarsening:** Selects $n_r$ random nodes from the graph and constructs the new adjacency matrix $\mathbf{A}_r$ via Equation (5).

(ii) **Topk Coarsening:** Selects $n_r$ nodes with the highest degrees before computing $\mathbf{A}_r$ using Equation (5). This approach may be computationally expensive for large graphs, as it requires full degree information.

(iii) **Subgraph Coarsening:** Samples a random node $j$ and selects its nearest $n_r$ neighbors. If geometric positions are available, the selection is based on Euclidean distance; otherwise, a $k - hop$ ego network is used. This method captures local graph structures but may overlook global properties.

(iv) **Hybrid Coarsening:** Alternates between Random and Subgraph Coarsening to leverage both local and global components of the graph.

Further details on the coarsening methods are provided in Appendix C.

**Experimental Settings.** We evaluate multiscale training with 2, 3, and 4 levels of coarsening, reducing the graph size by a factor of 2 at each level. Training epochs are doubled at each coarsening step, while fine-grid epochs remain fewer than in standard training. For multiscale gradient computation, we coarsen the graph to retain 75% of the nodes, perform half of the training using the coarsened graph, and then transition to fine-grid training. We validate our methods on node-level tasks, demonstrating that incorporating coarse levels during training enhances inference performance.

## 6.1 Transductive Learning

We evaluate our approach using the OGBN-Arxiv dataset (Hu et al., 2020) with GCN (Kipf & Welling, 2016), GIN Xu et al. (2018), and GAT (Velickovic et al., 2017). This dataset consists of 169,343 nodes and 1,166,243 edges, where nodes represent individual computer science papers, edges correspond to citation relationships where one paper cites another, and each node is labeled with the paper's primary subject area. We report test accuracy across three runs with different random seeds and include standard deviation to assess robustness, and present our results in Table 2, demonstrating that our multiscale framework achieves comparable or superior performance relative to standard training. Notably, random pooling proves to be the most effective approach for this dataset, offering both efficiency and strong performance. Given the large number of edges in this dataset, we use $p = 1$ and report the corresponding edge counts in Table 3. The results highlight the substantial reduction in edges at each pooling level, significantly improving training efficiency. In Table 2, we also provide a timing analysis, including both theoretical FLOP estimates (as elaborated in Section 5) and empirical runtime measurements, confirming that our multiscale approach enhances efficiency while maintaining strong predictive performance. To show the efficiency of our methods, we demonstrate a comparison of the convergence of the training loss between the multiscale training and fine grid training in Appendix D, and we report memory consumption measurements in Appendix E.

We demonstrate our results using the OGBN-MAG dataset (Hu et al., 2020), which is a heterogeneous network composed of a subset of the Microsoft Academic Graph (MAG). This dataset includes papers, authors, institutions, and fields of study, of which the papers are labeled. We focus on the 'paper' nodes and the 'citing' relation, using a homogeneous graph dataset of 736,389 nodes and 5,416,271 edges. We summarize our results in Table 4, where we show that using GCN (Kipf & Welling, 2016), our methods achieve results

Table 2: Comparison of different coarsening methods on the OGBN-Arxiv dataset. The lowest (best) FLOPs count for each GNN appears in blue. Time represents the training epoch time in milliseconds. The lowest (best) time for a given network and level is highlighted in green. The highest (best) performing approach in terms of test accuracy (%) is highlighted in **black**.

| Method | L = 2 FLOPs(M) | Time(ms) | Test Acc(%) | L = 3 FLOPs(M) | Time(ms) | Test Acc(%) | L = 4 FLOPs(M) | Time(ms) | Test Acc(%) |
|---|---|---|---|---|---|---|---|---|---|
| **GCN** | | | | | | | | | |
| Fine Grid ($L=1$) | | | | $1.96 \times 10^4$ | 39.70 | 71.76% (0.1%) | | | |
| Random | $9.37 \times 10^3$ | 26.65 | **72.07% (0.1%)** | $4.59 \times 10^3$ | 18.52 | **71.88% (0.0%)** | $2.27 \times 10^3$ | 15.53 | **72.03% (0.0%)** |
| Topk | $1.01 \times 10^4$ | 31.96 | 71.68% (0.0%) | $5.16 \times 10^3$ | 22.01 | 71.44% (0.2%) | $2.58 \times 10^3$ | 17.01 | 70.99% (0.2%) |
| Subgraphs | $1.61 \times 10^3$ | 53.71 | 71.06% (0.6%) | $8.06 \times 10^2$ | 53.27 | 71.18% (0.2%) | $4.05 \times 10^2$ | 54.46 | 71.13% (0.0%) |
| Rand. & Sub. | $5.49 \times 10^3$ | 37.44 | 70.71% (0.2%) | $2.70 \times 10^3$ | 34.50 | 70.93% (0.0%) | $1.34 \times 10^3$ | 33.77 | 70.53% (0.2%) |
| **GIN** | | | | | | | | | |
| Fine Grid ($L=1$) | | | | $3.75 \times 10^4$ | 61.66 | 68.39% (0.6%) | | | |
| Random | $1.86 \times 10^4$ | 28.96 | **69.21% (1.0%)** | $9.24 \times 10^3$ | 17.21 | **69.19% (4.8%)** | $4.61 \times 10^3$ | 12.23 | **69.30% (0.5%)** |
| Topk | $1.89 \times 10^4$ | 32.53 | 67.82% (3.2%) | $9.50 \times 10^3$ | 19.84 | 68.10% (1.6%) | $4.75 \times 10^3$ | 13.68 | 67.90% (1.0%) |
| Subgraphs | $2.52 \times 10^3$ | 47.47 | 67.13% (1.3%) | $1.26 \times 10^3$ | 47.10 | 66.46% (1.6%) | $6.30 \times 10^2$ | 47.69 | 66.03% (6.4%) |
| Rand. & Sub. | $1.06 \times 10^4$ | 39.56 | 65.87% (6.6%) | $5.25 \times 10^3$ | 31.16 | 64.79% (3.0%) | $2.62 \times 10^3$ | 32.36 | 67.46% (1.2%) |
| **GAT** | | | | | | | | | |
| Fine Grid ($L=1$) | | | | $1.13 \times 10^4$ | 54.71 | 71.07% (0.2%) | | | |
| Random | $5.32 \times 10^3$ | 30.32 | **71.70% (0.5%)** | $2.59 \times 10^3$ | 19.39 | **71.78% (0.0%)** | $1.28 \times 10^3$ | 16.48 | **71.12% (0.0%)** |
| Topk | $5.86 \times 10^3$ | 43.90 | 71.34% (0.3%) | $3.01 \times 10^3$ | 29.72 | 70.68% (0.1%) | $1.50 \times 10^3$ | 21.26 | 70.63% (0.1%) |
| Subgraphs | $5.04 \times 10^3$ | 52.96 | 70.24% (1.0%) | $2.52 \times 10^3$ | 50.72 | 70.10% (0.6%) | $1.26 \times 10^3$ | 55.15 | 69.82% (0.7%) |
| Rand. & Sub. | $5.18 \times 10^3$ | 43.05 | 70.09% (0.2%) | $2.56 \times 10^3$ | 35.79 | 69.62% (0.2%) | $1.27 \times 10^3$ | 35.33 | 70.06% (0.2%) |

Table 3: Edge number for coarsening strategies for the coarse-to-fine method, with the OGBN-Arxiv dataset. Fine grid edge number is in red.

| Method | 1st Coarsening | 2nd Coarsening | 3rd Coarsening |
|---|---|---|---|
| | Fine Grid: 1,166,243 | | |
| Random | 280,009 | 76,559 | 17,978 |
| Topk | 815,023 | 480,404 | 238,368 |

comparable to the baseline results. We demonstrate our results for 2, 3, and 4 levels of multiscale training, and note that even though GCN is widely used and may be applied to this dataset, other more advanced architectures may yield better results than the presented baseline. However, changing a chosen model will not harm the effectiveness of our proposed methods.

To emphasize that our method is general and will prove to be effective over various datasets and methods, we use additional datasets of various sizes and show that for every one of them, our methods prove to be efficient while yielding comparable or even better results compared to the original baseline. In Appendix F, we show results and details for the well-known Cora, SiteCeer, and PubMed datasets (Sen et al., 2008). We further test our methods on Flickr (Zeng et al., 2019), WikiCS (Mernyei & Cangea, 2020), DBLP (Bojchevski & Günnemanng, 2017), the transductive versions of Facebook, BlogCatalog, and PPI (Yang et al., 2020) datasets, detailed in Appendix G. We present results for Flickr and PPI, along with timing evaluations, as well as full results for all those datasets and additional experimental details. Additionally, in Appendix J, we provide ablation studies, where we study various coarsening ratios and epoch distribution.

## 6.2 Inductive Learning

We evaluate our method on the ShapeNet dataset (Chang et al., 2015), a point cloud dataset with multiple categories, for a node segmentation task. Specifically, we present results on the 'Airplane' category, which contains 2,690 point clouds, each consisting of 2,048 points. Since this dataset contains multiple graphs, its overall size is large, and achieving an improvement in training efficiency for each batch will yield a significant impact on training efficiency.

Table 4: Comparison of different methods for OGBN-MAG dataset. We present the test accuracy for multiscale training using GCN. Results for fine grid appear in red. $L$ represents the number of levels.

| Method ↓ / Levels → | $L = 2$ | $L = 3$ | $L = 4$ |
|---|---|---|---|
| | Fine grid: 35.73% (0.4%) | | |
| Random | 36.54% (0.5%) | 36.92% (0.1%) | 37.07% (0.2%) |
| Topk | 35.46% (0.4%) | 34.82% (0.3%) | 34.69% (0.2%) |
| Subgraphs | 34.19% (0.3%) | 34.31% (0.4%) | 34.31% (1.2%) |
| Rand. & Sub. | 34.64% (0.3%) | 34.43% (0.2%) | 34.05% (0.2%) |

Table 5: Results on Shapenet 'Airplane' dataset using DGCNN. Results represent the Mean test IoU. Results for fine grid appear in red.

| Training using Multiscale Training | | | | |
|---|---|---|---|---|
| Graph Density (kNN) | Fine Grid | Random | Subgraphs | Rand. & Sub. |
| k=6 | 78.36% (0.28%) | 79.86% (0.87%) | 79.31% (0.20%) | 80.01% (0.21%) |
| k=10 | 79.71% (0.23%) | 80.41% (0.14%) | 80.65% (0.15%) | 80.97% (0.10%) |
| k=20 | 81.06% (0.08%) | 81.37% (0.05%) | 81.18% (0.32%) | 81.89% (0.58%) |
| Training using Multiscale Gradient Computation Method | | | | |
| Graph Density (kNN) | Fine Grid | Random | Subgraphs | Rand. & Sub. |
| k=6 | 78.36% (0.28%) | 79.42% (1.40%) | 79.35% (0.12%) | 79.62% (0.93%) |
| k=10 | 79.71% (0.23%) | 80.00% (0.33%) | 80.15% (0.71%) | 80.19% (1.63%) |
| k=20 | 81.06% (0.08%) | 81.30% (0.41%) | 81.17% (0.25%) | 81.12% (0.24%) |

In this dataset, nodes are classified into four categories: fuselage, wings, tail, and engine. For training, we use Dynamic Graph Convolutional Neural Network (DGCNN) from Wang et al. (2018), which dynamically constructs graphs at each layer using the $k$-nearest neighbors algorithm, with $k$ as a tunable hyperparameter. We test our multiscale approach with random pooling, subgraph pooling, and a combination of both. In subgraph pooling, given that the data resides in $\mathbb{R}^3$, we construct subgraphs by selecting a random node and its $n_j$ nearest neighbors. Experiments are conducted using three random seeds, and we report the mean Intersection-over-Union (IoU) scores along with standard deviations in the top section of Table 5. Our results show that for all tested values of $k$, applying either random or subgraph pooling improves performance compared to standard training while reducing computational cost. Furthermore, combining both pooling techniques consistently yields the best results across connectivity settings, suggesting that this approach captures both high- and low-frequency components of the graph while maintaining training efficiency. Additional results for multiscale training on the full ShapeNet dataset are provided in Appendix H.

We apply the Multiscale Gradients Computation method to the ShapeNet 'Airplane' category, evaluating various pooling techniques using Algorithm 2 with two levels. As shown in the bottom section of Table 5, our approach reduces training costs while achieving performance comparable to fine-grid training. These results show that the early stages of optimization can be efficiently performed using an approximate loss, as we propose, with fine-grid calculations needed only once the loss function nears its minimum. Additional results, timing analyses and experimental details for the ShapeNet dataset are provided in Appendix H.

We further evaluate our Multiscale Gradients Computation method on the protein-protein interaction (PPI) dataset (Zitnik & Leskovec, 2017), consisting of graphs representing different human tissues. The results, with additional details, are provided in Appendix I. Furthermore, Appendix I also presents results on the NCI1 (Wale et al., 2008) and ogbg-molhiv (Hu et al., 2020) datasets.

## 7 Limitations

While our methods are broadly applicable across various models, datasets, and architectures, several limitations remain. Although our multiscale approach successfully avoids using the fine graph for most of the training process, a few iterations must still be conducted on the original fine grid, which requires it to fit in memory. Additionally, while the different variants of our method generally yield comparable performance, we currently lack a principled, deterministic criterion for selecting the optimal variant in a given setting, and specific cases may cause relatively high variance in certain datasets and methods. However, in this paper, we focus on the framework and the overall multiscale concept. Lastly, the theoretical analysis presented in this paper is restricted to the linear case (i.e., $\sigma = \mathrm{Id}$). We believe further theoretical investigation may be performed to extend the analysis to nonlinear settings.

## 8 Conclusions

In this work, we propose an efficient approach for training GNNs and evaluate its effectiveness across multiple datasets, architectures, and pooling strategies. We demonstrate that all proposed approaches, Coarse-to-Fine, Sub-to-Full, and Multiscale Gradient Computation, improve performance while reducing the training costs.

Our method is adaptable to any architecture, dataset, or pooling technique, making it a versatile tool for GNN training. Our experiments reveal that no single setting is universally optimal; performance varies depending on the dataset and architecture. However, our approach consistently yields improvements across all considered settings – for each dataset and network, at least one coarsening method led to improved performance, while the differences between methods were generally minor in practice.

## 9 Acknowledgements

This research was supported by the Ministry of Innovation, Science and Technology, Israel, and the Ministry of Education and Science of Ukraine as the Joint Ukrainian-Israeli research project (under contract M/38-2024, No. 0124U003512). Also, this research was partially supported by Grant No. 2023771 from the United States-Israel Binational Science Foundation (BSF) and by Grant No. 2411264 from the United States National Science Foundation (NSF).

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

## A  Theorem 4.1 Proof

Let us consider the model problem in Equation (11) for $\sigma = Id$

$$\mathcal{L}(\boldsymbol{\theta}) = \frac{1}{2N}\|\mathbf{AX}\boldsymbol{\theta} - \mathbf{Y}\|^2, \tag{15}$$

and the sketching problem from Woodruff (2014)

$$\mathcal{L}_C(\boldsymbol{\theta}) = \frac{1}{2N}\|\mathbf{P}^\top\mathbf{AX}\boldsymbol{\theta} - \mathbf{P}^\top\mathbf{Y}\|^2 \tag{16}$$

where $\mathbf{P}^\top$ is a sketching matrix.

Then the following theorem can be proved:

**Theorem A.1** (Woodruff (2014), Thm. 23). *Let $\mathbf{P}^\top$ be a sparse subspace embedding with $O(\frac{d^2}{\epsilon})$ rows, and let $\boldsymbol{\theta}^*$ be the solution of Equation (15) and $\boldsymbol{\theta}_C^*$ be the solution of Equation (16). Then, with probability 0.99, it holds that:*

$$\mathcal{L}_c(\boldsymbol{\theta}_C^*) = \frac{1}{2N}\|\mathbf{P}^\top\mathbf{AX}\boldsymbol{\theta}_C^* - \mathbf{P}^\top\mathbf{Y}\|^2 = \frac{1 \pm \epsilon}{2N}\|\mathbf{AX}\boldsymbol{\theta}^* - \mathbf{Y}\|^2 \tag{17}$$

In the sketching theory, the network is applied first, and only then is the graph size reduced. Here, we consider the operator $\mathbf{P}^\top$ that chooses a subset of nodes. Unlike sketching, in our multiscale training, we reduce the number of nodes first. We thus have that choosing a subset of nodes has a slightly different form, that is, our subgraph problem obeys

$$
\begin{aligned}
\mathcal{L}_c(\boldsymbol{\theta}_C^*) &= \frac{1}{2N}\|\mathbf{P}^\top\mathbf{APP}^\top\mathbf{X}\boldsymbol{\theta}_C^* - \mathbf{P}^\top\mathbf{Y}\|^2 = \frac{1}{2N}\|\mathbf{P}^\top\mathbf{AX}\boldsymbol{\theta}_C^* - \mathbf{P}^\top\mathbf{Y} + \mathbf{R}\|^2 \\
&\leq \frac{1}{2N}\left(\|\mathbf{P}^\top\mathbf{AX}\boldsymbol{\theta}_C^* - \mathbf{P}^\top\mathbf{Y}\|^2 + \|\mathbf{R}\|^2\right) = \frac{1 \pm \epsilon}{2N}\left(\|\mathbf{AX}\boldsymbol{\theta}^* - \mathbf{Y}\|^2 + \|\mathbf{R}\|^2\right).
\end{aligned}
\tag{18}
$$

Here we used norm properties and Theorem A.1, and $\mathbf{R}$ is the residual.

Now we focus on the case where $\mathbf{P}^\top$ chooses a subset of nodes. Assume that we reorder $\mathbf{A} = \begin{bmatrix} \mathbf{A}_{CC} & \mathbf{A}_{CF} \\ \mathbf{A}_{FC} & \mathbf{A}_{FF} \end{bmatrix}$, such that $C$ denotes the chosen node indices, and denote: $\mathbf{X} = \begin{bmatrix} \mathbf{X}_C \\ \mathbf{X}_F \end{bmatrix}$. Then, $\mathbf{P}^\top\mathbf{AP} = \mathbf{A}_{CC}$, and $\mathbf{P}^\top\mathbf{X} = \mathbf{X}_C$. We obtain that:

$$\mathbf{P}^\top\mathbf{APP}^\top\mathbf{X} = \mathbf{A}_{CC}\mathbf{X}_C$$

while

$$\mathbf{P}^\top\mathbf{AX} = \mathbf{A}_{CC}\mathbf{X}_C + \mathbf{A}_{CF}\mathbf{X}_F.$$

Hence, we have

$$\mathbf{P}^\top\mathbf{AX}\boldsymbol{\theta}_C^* = (\mathbf{A}_{CC}\mathbf{X}_C + \mathbf{A}_{CF}\mathbf{X}_F)\boldsymbol{\theta}_C^*$$

and

$$\mathbf{P}^\top\mathbf{AX}\boldsymbol{\theta}_C^* = \mathbf{A}_{CC}\mathbf{X}_C\boldsymbol{\theta}_C^* + \mathbf{A}_{CF}\mathbf{X}_F\boldsymbol{\theta}_C^* = \mathbf{P}^\top\mathbf{APP}^\top\mathbf{X}\boldsymbol{\theta}_C^* + \mathbf{A}_{CF}\mathbf{X}_F\boldsymbol{\theta}_C^*.$$

This means that $\|\mathbf{R}\| < \|\mathbf{A}_{CF}\mathbf{X}_F\boldsymbol{\theta}_C^*\|$. We conclude our statement with the following:

**Theorem A.2.** *Let $\boldsymbol{\theta}^*$ be the solution of equation (15) and let $\boldsymbol{\theta}_C^*$ be the solution of equation (16). Denote $\epsilon > 0$, and let $\mathbf{P}^\top$ be a sparse subspace embedding with $O(\frac{c^2}{\epsilon})$ rows, where $c$ is the size of the feature vectors (number of graph channels), that injects any vector into a subset of indices, which we denote by $C$. According to this subset, denote $\mathbf{A} = \begin{bmatrix} \mathbf{A}_{CC} & \mathbf{A}_{CF} \\ \mathbf{A}_{FC} & \mathbf{A}_{FF} \end{bmatrix}$, and $\mathbf{X} = \begin{bmatrix} \mathbf{X}_C \\ \mathbf{X}_F \end{bmatrix}$. Then if $\sigma = Id$, there exists $\|\mathbf{R}\| < \|\mathbf{A}_{CF}\mathbf{X}_F\boldsymbol{\theta}_C^*\|$ such that with probability .99:*

$$\mathcal{L}_c(\boldsymbol{\theta}_C^*) = \frac{1}{2N}\|\mathbf{P}^\top\mathbf{APP}^\top\mathbf{X}\boldsymbol{\theta}_C^* - \mathbf{P}^\top\mathbf{Y}\|^2 \leq \frac{1 \pm \epsilon}{2N}\left(\|\mathbf{AX}\boldsymbol{\theta}^* - \mathbf{Y}\|^2 + \|\mathbf{R}\|^2\right) \tag{19}$$

The approximation quality depends on the residual matrix $\mathbf{R}$. For separable graphs, there are no edges that are omitted and therefore $\mathbf{R} = 0$. As the number of edges that are omitted increases, the solution of the coarse-scale problem diverges from the solution of the fine-scale one. While theoretically the residual may become large in certain scenarios, our empirical results show that in practical settings $\mathbf{R}$ remains moderate.

When combining both coarsening and subgraphs, we have a diverse sampling of the graph, which may yield a better approximation on average, as can be shown for certain datasets. We present such datasets in our numerical experiments presented in B.

# B  Qtips and MNIST Datasets

We evaluate our methods on the Qtips dataset by constructing a training set of 100 rods and a test set of 1,000 rods. We employ a 4-layer GCN with 256 hidden channels and compare the performance of our approach across three levels of coarsening. The baseline is trained for 500 epochs, while the multiscale methods are trained for [400, 200, 100] epochs across levels. Results, shown in Table 6, indicate that the combination of random coarsening and subgraph pooling achieves the best performance, particularly when using $p = 3$.

We further test our approach on a graph dataset derived from the MNIST image dataset (LeCun et al., 1998). The original MNIST images consist of $28 \times 28$ grayscale pixels, each depicting a digit from 0 to 9. We convert each image into a graph by treating each pixel as a node and connecting nodes using their spatial positions and a $k$-nearest-neighbor algorithm. The task is to assign each node either to a background class or to the digit class it belongs to. We train a 4-layer GCN with 256 hidden channels on 1,000 graphs and evaluate on 100 test graphs. Results are presented in Table 7, where we again observe that the combination of random coarsening and subgraph pooling achieves the best performance among the methods evaluated.

Table 6: Qtips dataset results, comparing the baseline with 3 levels of coarsening.

|     | Random | Topk | Subgraphs | Random and Subgraphs |
|-----|--------|------|-----------|----------------------|
| **Fine Grid:** 73.03% (0.1%) | | | | |
| p=1 | 71.47% (0.2%) | 71.46% (0.1%) | 72.97% (0.1%) | 73.20% (0.0%) |
| p=2 | 71.40% (0.1%) | 71.30% (0.1%) | 72.82% (0.1%) | 73.02% (0.1%) |
| p=3 | 71.31% (0.1%) | 71.30% (0.1%) | 72.72% (0.1%) | **73.30% (0.1%)** |

Table 7: MNIST dataset results, comparing the baseline with 3 levels of coarsening.

|     | Random | Topk | Subgraphs | Random and Subgraphs |
|-----|--------|------|-----------|----------------------|
| **Fine Grid:** 86.28% (0.1%) | | | | |
| p=1 | 85.78% (0.1%) | 85.85% (0.0%) | 85.98% (0.0%) | 85.96% (0.0%) |
| p=2 | 85.77% (0.1%) | 85.88% (0.1% | 85.97% (0.1%) | 85.99% (0.0%) |
| p=3 | 85.85% (0.1%) | 85.90% (0.1%) | 86.03% (0.0%) | **86.04% (0.1%)** |

## C   Coarsening Methods

Additional implementation details of our coarsening procedures are provided. Algorithm 3 presents the random coarsening algorithm, Algorithm 4 outlines the Top-$k$ coarsening method, and Algorithm 5 describes the subgraph generation process.

---

**Algorithm 3** Random Coarsening

---

$\# \ m$ – *coarsining ratio.*
Initialize Adj $= \mathbf{A}$
**for** $i = 0, \ldots, p - 1$ **do**
    $\mathbf{A} = \mathbf{A} \times$ Adj
**end for**
Randomly choose m nodes
Use chosen nodes to calculate the injection matrix $\mathbf{P}$
Use $\mathbf{P}$ to calculate $\mathbf{A}_r$ as in Equation (5)

---

**Algorithm 4** Topk Coarsening

---

$\# \ m$ – *coarsining ratio.*
Initialize Adj $= \mathbf{A}$
**for** $i = 0, \ldots, p - 1$ **do**
    $\mathbf{A} = \mathbf{A} \times$ Adj
**end for**
Select $m$ nodes with the largest degrees
Use chosen nodes to calculate the injection matrix $\mathbf{P}$
Use $\mathbf{P}$ to calculate $\mathbf{A}_r$ as in Equation (5)

---

**Algorithm 5** Subgraph Coarsening

---

$\# \ m$ – *coarsining ratio.*
$\# \ k_r$ – $k - hop$ *parameter for each level* $r$.
Initialize Adj $= \mathbf{A}$
**for** $i = 0, \ldots, p - 1$ **do**
    $\mathbf{A} = \mathbf{A} \times$ Adj
**end for**
Choose a random node
Generate subgraph using $k_r - hops$
Use chosen nodes to calculate the injection matrix $\mathbf{P}$
Use $\mathbf{P}$ to calculate $\mathbf{A}_r$ as in Equation (5)

---

# D   Loss Comparison between Multilevel Training and Single-Level Training

To further demonstrate the efficiency of our methods, we show in Figure 5 a comparison between the loss during training, both using single-level training and multiscale training. We show that both training approaches converge to similar loss after a similar number of training epochs, while as demonstrated in Table 2, the cost of a single iteration on the coarse graph is lower than on the original fine graph. The plot shows results for the OGBN-Arxiv dataset (Hu et al., 2020), using GCN (Kipf & Welling, 2016) and random pooling.

Table 8 presents the $\gamma_r$ values obtained during multiscale training, computed according to Equation 9. We note that $\gamma_r$ increases as the value of $r$ grows.

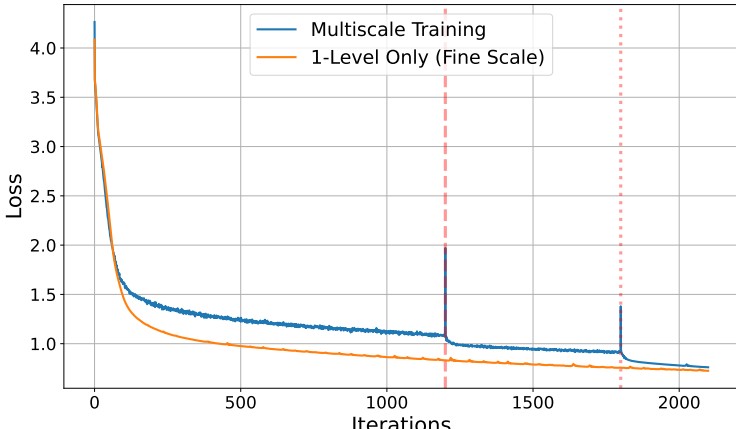

Figure 5: Comparison between losses using multiscale and single-level training. Coarse levels are generated using random coarsening.

Table 8: Demonstration of $\gamma_r$ values, as described in Equation 9.

| $\gamma_r$ | $r=1$ | $r=2$ | $r=3$ |
|---|---|---|---|
| Random | 0 | 0.060 | 0.076 |
| Topk | 0 | 0.046 | 0.175 |
| Subgraph | 0 | 0.067 | 0.170 |

# E   Memory Consumption

We show memory consumptions for OGBN-Arxiv (Hu et al., 2020), as function of coarsening layer and network. The results are summarized in Table 9, and they show that coarse grids consume less memory than the original fine scale data.

Table 9: Memory consumption across layers and models.

|  | $L = 2$ | $L = 3$ | $L = 4$ |
|---|---|---|---|
| **GCN** | | | |
| Fine Grid ($L = 1$) | | 1610.30 | |
| Random | 926.62 | 528.33 | 332.99 |
| Topk | 994.36 | 582.51 | 366.58 |
| Subgraphs | 641.59 | 491.47 | 221.85 |
| Rand. & Sub. | 923.39 | 528.28 | 333.20 |
| **GIN** | | | |
| Fine Grid ($L = 1$) | | 2188.64 | |
| Random | 1227.56 | 719.56 | 431.26 |
| Topk | 1259.72 | 739.62 | 443.28 |
| Subgraphs | 938.48 | 498.18 | 224.91 |
| Rand. & Sub. | 1228.66 | 718.55 | 431.26 |
| **GAT** | | | |
| Fine Grid ($L = 1$) | | 4059.53 | |
| Random | 1415.95 | 588.60 | 324.92 |
| Topk | 2787.29 | 1655.82 | 896.41 |
| Subgraphs | 1108.56 | 466.31 | 220.81 |
| Rand. & Sub. | 1423.09 | 601.40 | 322.41 |

## F    Cora, CiteSeer and PubMed Datasets

We evaluate our method on the Cora, Citeseer, and PubMed datasets (Sen et al., 2008) using the standard split from Yang et al. (2016), with 20 nodes per class for training, 500 validation nodes, and 1,000 testing nodes. Cora contains machine learning research papers classified into topics, CiteSeer includes scientific publications from various computer science fields, and PubMed consists of biomedical papers related to diabetes, where nodes represent documents and edges represent citation links. Our experiments cover standard architectures—GCN (Kipf & Welling, 2016), GIN (Xu et al., 2018), and GAT (Velickovic et al., 2017)—and are summarized in Table 10, where we achieve performance that is either superior to or on par with the fine-grid baseline, which trains directly on the original graphs.

We explore two pooling strategies—random and Top-k—along with subgraph generation via $k$-hop neighborhoods and a hybrid approach combining both. Each method is tested under two connectivity settings: the original adjacency ($p = 1$) and an enhanced version ($p = 2$), where we square the adjacency matrix. We report test accuracy across three runs with different random seeds and include standard deviation to assess robustness. Our results indicate that enhanced connectivity improves accuracy in some settings. Table 10 further presents three levels of multiscale training alongside theoretical FLOP analyses, demonstrating the efficiency of our approach. The details of the FLOPs analysis are provided in Section 5, and additional results for two and four levels of multiscale training are provided in Table 11 and Table 12. We provide further experimental details in Table 18, and additional information about the datasets appears in Table 14.

Table 10: Comparison of different coarsening methods on the Cora, CiteSeer, and PubMed datasets. The lowest (best) FLOPs count for each GNN appears in blue. The highest (best) performing approach in terms of test accuracy (%) is highlighted in **black**.

| Method ↓ / Dataset → | Cora | | CiteSeer | | PubMed | |
|---|---|---|---|---|---|---|
| | FLOPs(M) | Test Acc(%) | FLOPs(M) | Test Acc(%) | FLOPs(M) | Test Acc(%) |
| **GCN** | | | | | | |
| Fine grid | $9.91 \times 10^2$ | 82.9% (1.1%) | $2.69 \times 10^3$ | 69.6% (1.0%) | $3.55 \times 10^3$ | 77.5% (0.4%) |
| Random(p=1) | $2.40 \times 10^2$ | 82.0% (0.5%) | $6.58 \times 10^2$ | 70.4% (0.2%) | $8.51 \times 10^2$ | 77.5% (0.6%) |
| Random(p=2) | $2.63 \times 10^2$ | 83.0% (0.5%) | $6.82 \times 10^2$ | 66.0% (0.8%) | $9.99 \times 10^2$ | 73.8% (0.4%) |
| Topk(p=1) | $2.49 \times 10^2$ | 79.9% (4.4%) | $6.83 \times 10^2$ | 68.1% (0.1%) | $9.33 \times 10^2$ | **78.2% (0.3%)** |
| Topk(p=2) | $3.09 \times 10^2$ | 79.6% (2.4%) | $8.08 \times 10^2$ | 63.6% (2.7%) | $1.87 \times 10^3$ | 74.5% (0.4%) |
| Subgraphs(p=1) | $2.43 \times 10^2$ | **83.5% (0.4%)** | $6.56 \times 10^2$ | **71.2% (0.4%)** | $8.63 \times 10^2$ | 78.0% (0.3%) |
| Subgraphs(p=2) | $3.26 \times 10^2$ | 83.1% (0.5%) | $6.82 \times 10^2$ | 70.9% (0.2%) | $1.19 \times 10^3$ | 76.1% (0.3%) |
| Rand. & Sub.(p=1) | $2.41 \times 10^2$ | 81.5% (0.3%) | $6.57 \times 10^2$ | 70.0% (0.0%) | $8.57 \times 10^2$ | 77.8% (0.8%) |
| Rand. & Sub.(p=2) | $2.92 \times 10^2$ | 82.6% (0.5%) | $6.83 \times 10^2$ | 70.9% (0.2%) | $1.07 \times 10^3$ | 73.9% (0.4%) |
| **GIN** | | | | | | |
| Fine grid | $2.37 \times 10^3$ | 76.4% (1.6%) | $6.79 \times 10^3$ | 63.8% (2.6%) | $7.75 \times 10^3$ | 75.4% (0.4%) |
| Random(p=1) | $5.89 \times 10^2$ | 76.2% (2.7%) | $1.69 \times 10^3$ | 66.5% (2.2%) | $1.92 \times 10^3$ | 75.7% (2.0%) |
| Random(p=2) | $6.00 \times 10^2$ | 77.2% (0.6%) | $1.70 \times 10^3$ | 60.5% (0.8%) | $1.99 \times 10^3$ | 75.1% (0.9%) |
| Topk(p=1) | $5.94 \times 10^2$ | 77.6% (5.6%) | $1.70 \times 10^3$ | 64.8% (3.9%) | $1.96 \times 10^3$ | 75.6% (1.6%) |
| Topk(p=2) | $6.22 \times 10^2$ | 74.6% (3.0%) | $1.76 \times 10^3$ | 60.5% (2.0%) | $2.40 \times 10^3$ | 76.5% (1.5%) |
| Subgraphs(p=1) | $5.91 \times 10^2$ | **78.7% (1.1%)** | $1.69 \times 10^3$ | **68.8% (1.6%)** | $1.93 \times 10^3$ | **77.2% (1.2%)** |
| Subgraphs(p=2) | $6.31 \times 10^2$ | 76.9% (1.72%) | $1.70 \times 10^3$ | 65.0% (1.8%) | $2.08 \times 10^3$ | 75.2% (1.5%) |
| Rand. & Sub.(p=1) | $5.90 \times 10^2$ | 76.9% (1.0%) | $1.69 \times 10^3$ | 65.3% (1.2%) | $1.92 \times 10^3$ | 76.1% (2.8%) |
| Rand. & Sub.(p=2) | $6.14 \times 10^2$ | 75.9% (0.8%) | $1.70 \times 10^3$ | 66.0% (0.2%) | $2.02 \times 10^3$ | 75.9% (0.5%) |
| **GAT** | | | | | | |
| Fine grid | $1.11 \times 10^3$ | 79.6% (2.05%) | $3.35 \times 10^3$ | 67.2% (6.5%) | $3.08 \times 10^3$ | 75.5% (0.5%) |
| Random(p=1) | $2.65 \times 10^2$ | 80.3% (0.2%) | $8.11 \times 10^2$ | **72.9% (1.5%)** | $7.29 \times 10^2$ | 77.5% (1.5%) |
| Random(p=2) | $3.01 \times 10^2$ | 79.5% (0.6%) | $8.55 \times 10^2$ | 70.7% (1.0%) | $9.01 \times 10^2$ | 77.4% (1.2%) |
| Topk(p=1) | $2.79 \times 10^2$ | 79.8% (0.6%) | $8.56 \times 10^2$ | 66.8% (0.6%) | $8.24 \times 10^2$ | 76.1% (0.9%) |
| Topk(p=2) | $3.72 \times 10^2$ | 76.8% (6.7%) | $1.08 \times 10^3$ | 65.2% (0.7%) | $1.92 \times 10^3$ | 74.9% (0.7%) |
| Subgraphs(p=1) | $2.70 \times 10^2$ | 78.5% (0.8%) | $8.08 \times 10^2$ | 70.2% (0.6%) | $7.43 \times 10^2$ | 75.4% (4.6%) |
| Subgraphs(p=2) | $3.99 \times 10^2$ | **80.7% (1.2%)** | $8.54 \times 10^2$ | 68.6% (0.3%) | $1.12 \times 10^3$ | 77.3% (1.1%) |
| Rand. & Sub.(p=1) | $2.67 \times 10^2$ | 79.9% (1.4%) | $8.10 \times 10^2$ | 71.2% (0.8%) | $7.36 \times 10^2$ | 76.8% (0.5%) |
| Rand. & Sub.(p=2) | $3.46 \times 10^2$ | 80.2% (1.1%) | $8.57 \times 10^2$ | 70.8% (0.4%) | $9.87 \times 10^2$ | **77.5% (1.1%)** |

Table 11: Comparison of different coarsening methods on the Cora and CiteSeer datasets. The highest (best) performing approach in terms of test accuracy (%) is highlighted in **black**.

| Method ↓ / Dataset → | Cora | | | CiteSeer | | |
|---|---|---|---|---|---|---|
| | $L = 2$ | $L = 3$ | $L = 4$ | $L = 2$ | $L = 3$ | $L = 4$ |
| **GCN** | | | | | | |
| Fine Grid ($L = 1$) | | 82.9% (1.1%) | | | 69.6% (1.0%) | |
| Random(p=1) | 81.7% (0.6%) | 82.0% (0.5%) | 81.7% (0.4%) | 70.5% (0.3%) | 70.4% (0.2%) | 70.7% (0.2%) |
| Random(p=2) | **83.2% (0.6%)** | 83.0% (0.5%) | 81.5% (0.3%) | 66.6% (2.6%) | 66.0% (0.8%) | 66.2% (2.2%) |
| Topk(p=1) | 79.5% (0.7%) | 79.9% (4.4%) | 79.8% (2.2%) | 68.5% (1.1%) | 68.1% (0.1%) | 67.4% (0.3%) |
| Topk(p=2) | 80.5% (2.5%) | 79.6% (2.4%) | 80.8% (0.8%) | 66.4% (0.8%) | 63.6% (2.7%) | 65.8% (2.1%) |
| Subgraphs(p=1) | 82.7% (0.7%) | **83.5% (0.4%)** | 81.3% (0.1%) | **71.6% (0.5%)** | **71.2% (0.4%)** | **71.3% (0.4%)** |
| Subgraphs(p=2) | 82.8% (0.4%) | 83.1% (0.5%) | **81.8% (0.7%)** | 71.5% (0.5%) | 70.9% (0.2%) | 70.9% (0.4%) |
| Rand. & Sub.(p=1) | 81.7% (0.4%) | 81.5% (0.3%) | 81.5% (0.3%) | 71.4% (0.5%) | 70.0% (0.0%) | 70.7% (0.3%) |
| Rand. & Sub.(p=2) | 82.7% (0.3%) | 82.6% (0.5%) | 80.8% (0.1%) | 70.8% (0.3%) | 70.9% (0.2%) | 70.7% (0.1%) |
| **GIN** | | | | | | |
| Fine Grid ($L = 1$) | | 76.4% (1.6%) | | | 63.8% (2.6%) | |
| Random(p=1) | 76.6% (1.5%) | 76.2% (2.7%) | 77.6% (2.2%) | 66.3% (7.0%) | 66.5% (2.2%) | 66.4% (1.1%) |
| Random(p=2) | 73.1% (0.3%) | 77.2% (0.6%) | 71.6% (1.6%) | 60.6% (2.6%) | 60.5% (0.8%) | 62.8% (1.3%) |
| Topk(p=1) | 77.5% (1.3%) | 77.6% (5.6%) | **79.4% (1.9%)** | 67.3% (1.3%) | 64.8% (3.9%) | 66.5% (1.5%) |
| Topk(p=2) | 76.2% (1.6%) | 74.6% (3.0%) | 75.6% (2.3%) | 61.1% (1.1%) | 60.5% (2.0%) | 60.0% (2.9%) |
| Subgraphs(p=1) | 77.5% (0.9%) | **78.7% (1.1%)** | 78.2% (1.4%) | **68.2% (0.5%)** | **68.8% (1.6%)** | **68.7% (1.4%)** |
| Subgraphs(p=2) | 76.2% (8.57%) | 76.9% (1.72%) | 77.3% (8.55%) | 60.1% (0.29%) | 65.0% (1.8%) | 66.6% (1.0%) |
| Rand. & Sub.(p=1) | **80.2% (2.5%)** | 76.9% (1.0%) | 76.8% (2.1%) | 66.0% (0.9%) | 65.3% (1.2%) | 65.5% (1.1%) |
| Rand. & Sub.(p=2) | 75.5% (0.7%) | 75.9% (0.8%) | 74.6% (22.5%) | 64.1% (2.2%) | 66.0% (0.2%) | 66.7% (1.0%) |
| **GAT** | | | | | | |
| Fine Grid ($L = 1$) | | 79.6% (2.05%) | | | 67.2% (6.5%) | |
| Random(p=1) | 80.1% (0.8%) | 80.3% (0.2%) | 79.8% (0.8%) | 71.1% (0.4%) | **72.9% (1.5%)** | **71.4% (0.3%)** |
| Random(p=2) | **82.0% (0.5%)** | 79.5% (0.6%) | 79.5% (1.9%) | **71.6% (1.1%)** | 70.7% (1.0%) | 70.6% (1.5%) |
| Topk(p=1) | 77.3% (1.8%) | 79.8% (0.6%) | 77.5% (2.9%) | 67.2% (1.5%) | 66.8% (0.6%) | 70.8% (1.3%) |
| Topk(p=2) | 79.6% (2.3%) | 76.8% (6.7%) | 79.0% (1.6%) | 67.4% (2.2%) | 65.2% (0.7%) | 67.0% (0.2%) |
| Subgraphs(p=1) | 80.2% (2.6%) | 78.5% (0.8%) | 79.5% (0.8%) | 69.1% (0.6%) | 70.2% (0.6%) | 68.4% (2.7%) |
| Subgraphs(p=2) | 80.5% (1.5%) | **80.7% (1.2%)** | 79.9% (0.9%) | 70.1% (0.5%) | 68.6% (0.3%) | 66.0% (0.2%) |
| Rand. & Sub.(p=1) | 80.6% (0.2%) | 79.9% (1.4%) | **80.3% (0.9%)** | 70.8% (0.1%) | 71.2% (0.8%) | 71.2% (0.3%) |
| Rand. & Sub.(p=2) | 81.1% (1.1%) | 80.2% (1.1%) | 78.7% (0.4%) | 70.3% (0.4%) | 70.8% (0.4%) | 71.1% (0.3%) |

Table 12: Comparison of different coarsening methods on the PubMed dataset. The highest (best) performing approach in terms of test accuracy (%) is highlighted in **black**.

| Method ↓ / Levels → | $L = 2$ | $L = 3$ | $L = 4$ |
|---|---|---|---|
| **GCN** | | | |
| Fine Grid ($L = 1$) | | 77.5% (0.4%) | |
| Random(p=1) | 77.6% (0.8%) | 77.5% (0.6%) | 77.5% (0.4%) |
| Random(p=2) | 74.2% (0.4%) | 73.8% (0.4%) | 74.2% (0.7%) |
| Topk(p=1) | **77.8% (0.9%)** | **78.2% (0.3%)** | 78.2% (0.6%) |
| Topk(p=2) | 75.4% (0.4%) | 74.5% (0.4%) | 74.2% (0.6%) |
| Subgraphs(p=1) | 77.7% (0.2%) | 78.0% (0.3%) | **78.7% (1.3%)** |
| Subgraphs(p=2) | 74.8% (1.2%) | 76.1% (0.3%) | 76.5% (1.1%) |
| Rand. & Sub.(p=1) | 77.4% (0.5%) | 77.8% (0.8%) | 76.7% (0.3%) |
| Rand. & Sub.(p=2) | 74.1% (0.7%) | 73.9% (0.4%) | 75.6% (0.9%) |
| **GIN** | | | |
| Fine Grid ($L = 1$) | | 75.4% (0.4%) | |
| Random(p=1) | 75.6% (1.7%) | 75.7% (2.0%) | 76.2% (1.3%) |
| Random(p=2) | 75.6% (2.9%) | 75.1% (0.9%) | 74.3% (1.2%) |
| Topk(p=1) | 76.6% (0.8%) | 75.6% (1.6%) | 75.5% (1.6%) |
| Topk(p=2) | 76.0% (1.8%) | 76.5% (1.5%) | 66.1% (6.4%) |
| Subgraphs(p=1) | 75.8% (1.2%) | **77.2% (1.2%)** | **77.2% (0.7%)** |
| Subgraphs(p=2) | 75.6% (2.8%) | 75.2% (1.5%) | 55.4% (4.7%) |
| Rand. & Sub.(p=1) | 75.2% (0.8%) | 76.1% (2.8%) | 75.0% (0.7%) |
| Rand. & Sub.(p=2) | **77.8% (5.6%)** | 75.9% (0.5%) | 76.4% (2.4%) |
| **GAT** | | | |
| Fine Grid ($L = 1$) | | 75.5% (0.5%) | |
| Random(p=1) | 76.5% (0.3%) | 77.5% (1.5%) | 76.4% (0.8%) |
| Random(p=2) | 75.2% (1.0%) | 77.4% (1.2%) | **78.6% (2.5%)** |
| Topk(p=1) | 76.1% (0.2%) | 76.1% (0.9%) | 75.6% (2.5%) |
| Topk(p=2) | 74.2% (1.0%) | 74.9% (0.7%) | 73.3% (1.0%) |
| Subgraphs(p=1) | 77.0% (1.0%) | 75.4% (4.6%) | 75.8% (1.6%) |
| Subgraphs(p=2) | 75.1% (0.85%) | 77.3% (1.1%) | 78.2% (3.0%) |
| Rand. & Sub.(p=1) | **77.3% (0.8%)** | 76.8% (0.5%) | 76.0% (1.9%) |
| Rand. & Sub.(p=2) | 75.7% (1.8%) | **77.5% (1.1%)** | 76.4% (1.9%) |

# G   Additional Transductive Learning Datasets

We further test our methods using additional datasets. We provide results for Flickr Zeng et al. (2019) and PPI (transductive) Yang et al. (2020), showing both timing analysis, in Table 13, and full multiscale training results in Table 15. Flickr is a social network where nodes represent images and edges represent mutual user connections or shared tags, focusing on image-based community detection, while PPI (transductive) is a protein-protein interaction network where nodes represent proteins and edges denote biological interactions, with the goal of predicting protein functions within a single connected graph. While pooling does not produce a clearly interpretable or meaningful molecule in the molecular setting, this does not affect the applicability or performance of our methods. Our results show that while achieving comparable accuracy to the original fine grid, our methods show significant timing improvement both in a theoretical manner as well as in FLOPs analysis, which we calculate as explained in Section 5.

In addition, we test our methods and show results for Facebook, BlogCatalog (Yang et al., 2020), DBLP (Bojchevski & Günnemanng, 2017) and WikiCS (Mernyei & Cangea, 2020), in Table 16 and Table 17. Facebook represents a social network where nodes are users and edges represent friendships, BlogCatalog is a social network of bloggers connected based on shared interests, DBLP is a co-authorship network built from academic publications in computer science, and WikiCS consists of Wikipedia articles about computer science topics connected by hyperlinks. This further illustrates that our methods are not restricted to a specific dataset or network architecture, as our results are consistently comparable to the original training baseline results. Details of our experiments are provided in Table 18, and we list additional details of the datasets in Table 14.

We provide additional comparison between our results and sampling-based efficient GNN training methods, which include GraphSAINT (Zeng et al., 2020), FastGCN (Chen et al., 2018), and ClusterGCN (Chiang et al., 2019). Table 19 shows a comparison with our methods, for the Flickr (Zeng et al., 2019) dataset. We observe that our efficient methods have yielded better results for this dataset in comparison to the above methods.

Table 13: Comparison of different coarsening methods on the Flickr and PPI (transductive) datasets. The lowest (best) FLOPs count for each GNN appears in blue. Time represents the training epoch time in milliseconds. The lowest (best) time for a given network and level is highlighted in green. The highest (best) performing approach in terms of test accuracy (%) is highlighted in **black**.

| Method ↓ / Dataset → | Flickr | | | PPI (transductive) | | |
|---|---|---|---|---|---|---|
| | FLOPs(M) | Time(ms) | Test Acc(%) | FLOPs(M) | Time(ms) | Test Acc(%) |
| **GCN** | | | | | | |
| Fine grid | $1.72 \times 10^4$ | 68.90 | 54.77% (0.2%) | $8.09 \times 10^3$ | 64.66 | 70.74% (0.2%) |
| Random | $3.94 \times 10^3$ | 27.73 | 55.30% (1.6%) | $1.65 \times 10^3$ | 25.60 | 71.10% (0.2%) |
| Topk | $4.28 \times 10^3$ | 36.31 | 55.14% (0.1%) | $2.46 \times 10^3$ | 46.14 | 71.37% (0.4%) |
| Subgraphs | $4.92 \times 10^3$ | 73.16 | 54.61% (0.8%) | $1.61 \times 10^3$ | 113.90 | **71.63% (0.3%)** |
| Rand. & Sub. | $4.44 \times 10^3$ | 49.74 | **55.85% (1.6%)** | $1.63 \times 10^3$ | 68.06 | 71.50% (0.0%) |
| **GIN** | | | | | | |
| Fine grid | $3.58 \times 10^4$ | 76.29 | 52.05% (1.0%) | $1.34 \times 10^4$ | 48.76 | 85.67% (7.2%) |
| Random | $8.77 \times 10^3$ | 27.93 | 52.16% (3.9%) | $3.17 \times 10^3$ | 21.47 | 89.24% (0.4%) |
| Topk | $8.93 \times 10^3$ | 34.47 | 51.99% (5.1%) | $3.54 \times 10^3$ | 33.98 | 84.92% (0.8%) |
| Subgraphs | $9.23 \times 10^3$ | 75.88 | **53.41% (1.0%)** | $3.15 \times 10^3$ | 117.70 | 87.40% (8.1%) |
| Rand. & Sub. | $9.01 \times 10^3$ | 53.51 | 52.00% (1.8%) | $3.16 \times 10^3$ | 68.52 | **90.83% (12.3%)** |
| **GAT** | | | | | | |
| Fine grid | $1.53 \times 10^4$ | 103.54 | 52.71% (0.9%) | $4.57 \times 10^3$ | 192.20 | 82.37% (3.1%) |
| Random | $3.40 \times 10^3$ | 29.57 | 52.91% (0.3%) | $9.32 \times 10^2$ | 32.30 | 82.93% (1.3%) |
| Topk | $3.80 \times 10^3$ | 44.35 | 52.79% (0.4%) | $1.39 \times 10^3$ | 99.79 | 73.13% (1.6%) |
| Subgraphs | $4.55 \times 10^3$ | 94.21 | **53.37% (0.1%)** | $9.10 \times 10^2$ | 132.67 | **87.71% (1.8%)** |
| Rand. & Sub. | $3.99 \times 10^3$ | 62.38 | 53.26% (0.2%) | $9.21 \times 10^2$ | 79.21 | 87.03% (0.8%) |

Table 14: Datasets statistics.

| Dataset | Nodes | Edges | Features | Classes |
|---|---|---|---|---|
| Cora Sen et al. (2008) | 2,708 | 10,556 | 1,433 | 7 |
| Citeseer Sen et al. (2008) | 3,327 | 9,104 | 3,703 | 6 |
| Pubmed Sen et al. (2008) | 19,717 | 88,648 | 500 | 3 |
| Flickr Zeng et al. (2019) | 89,250 | 899,756 | 500 | 7 |
| WikiCS Mernyei & Cangea (2020) | 11,701 | 216,123 | 300 | 10 |
| DBLP Bojchevski & Günnemanng (2017) | 17,716 | 105,734 | 1,639 | 4 |
| Facebook Yang et al. (2020) | 4,039 | 88,234 | 1,283 | 193 |
| BlogCatalog Yang et al. (2020) | 5,196 | 343,486 | 8,189 | 6 |
| PPI (transductive) Yang et al. (2020) | 56,944 | 1,612,348 | 50 | 121 |

Table 15: Comparison of different coarsening methods on the Flickr and PPI (transductive) datasets. The highest (best) performing approach in terms of test accuracy (%) is highlighted in **black**.

| Method ↓ / Dataset → | Flickr | | | PPI (transductive) | | |
|---|---|---|---|---|---|---|
| | $L = 2$ | $L = 3$ | $L = 4$ | $L = 2$ | $L = 3$ | $L = 4$ |
| **GCN** | | | | | | |
| Fine Grid ($L = 1$) | | 54.77% (0.2%) | | | 70.74% (0.2%) | |
| Random | 55.01% (0.6%) | 55.30% (1.6%) | 54.47% (0.5%) | **70.97% (0.2%)** | 71.10% (0.2%) | 71.44% (0.1%) |
| Topk | 55.28% (0.2%) | 55.14% (0.1%) | **55.15% (0.1%)** | 70.62% (0.0%) | 71.37% (0.4%) | 71.09% (0.3%) |
| Subgraphs | 55.25% (0.1%) | 54.61% (0.8%) | 55.47% (0.3%) | 70.96% (0.1%) | **71.63% (0.3%)** | **73.07% (0.4%)** |
| Rand. & Sub.s | **55.65% (0.1%)** | **55.85% (1.6%)** | 55.08% (0.2%) | 70.88% (0.1%) | 71.50% (0.0%) | 72.89% (0.1%) |
| **GIN** | | | | | | |
| Fine Grid ($L = 1$) | | 52.05% (1.0%) | | | 85.67% (7.2%) | |
| Random | 52.24% (0.6%) | 52.16% (3.9%) | **53.74% (4.2%)** | 84.85% (2.6%) | 89.24% (0.4%) | 89.43% (0.6%) |
| Topk | **53.14% (3.5%)** | 51.99% (5.1%) | 52.19% (5.3%) | 81.74% (1.2%) | 84.92% (0.8%) | 83.70% (1.2%) |
| Subgraphs | 53.09% (1.4%) | **53.41% (1.0%)** | 53.33% (2.5%) | **91.27% (8.8%)** | 87.40% (8.1%) | 88.58% (0.2%) |
| Rand. & Sub. | 52.74% (1.0%) | 52.00% (1.8%) | 52.50% (2.0%) | 84.25% (4.1%) | **90.83% (12.3%)** | **93.31% (12.1%)** |
| **GAT** | | | | | | |
| Fine Grid ($L = 1$) | | 52.71% (0.9%) | | | 82.37% (3.1%) | |
| Random | 52.98% (0.3%) | 52.91% (0.3%) | 52.18% (1.5%) | 81.23% (1.4%) | 82.93% (1.3%) | 82.97% (3.0%) |
| Topk | **52.98% (0.2%)** | 52.79% (0.4%) | 52.72% (0.2%) | 77.39% (1.4%) | 73.13% (1.6%) | 70.37% (0.4%) |
| Subgraphs | 52.96% (0.3%) | **53.37% (0.1%)** | **53.33% (0.3%)** | **84.99% (0.9%)** | **87.71% (1.8%)** | 86.26% (1.9%) |
| Rand. & Sub. | 52.70% (0.0%) | 53.26% (0.2%) | 52.97% (0.2%) | 84.48% (1.0%) | 87.03% (0.8%) | **87.15% (1.3%)** |

Table 17: Comparison of different coarsening methods on the BlogCatalog and WikiCS datasets. The highest (best) performing approach in terms of test accuracy (%) is highlighted in **black**.

| Method ↓ / Dataset → | BlogCatalog | | | WikiCS | | |
|---|---|---|---|---|---|---|
| | $L=2$ | $L=3$ | $L=4$ | $L=2$ | $L=3$ | $L=4$ |
| **GCN** | | | | | | |
| Fine Grid ($L=1$) | | 75.87% (0.4%) | | | 78.07% (0.0%) | |
| Random | 76.54% (0.3%) | 76.63% (0.2%) | 76.54% (0.3%) | **79.55% (0.1%)** | 78.35% (0.4%) | **78.96% (0.1%)** |
| Topk | 76.44% (0.1%) | 76.25% (0.3%) | 75.96% (0.3%) | 77.95% (0.4%) | 78.06% (0.2%) | 77.83% (0.2%) |
| Subgraphs | **76.73% (0.1%)** | **77.88% (0.3%)** | **77.40% (0.3%)** | 78.36% (0.3%) | 78.28% (0.1%) | 78.42% (0.1%) |
| Rand. & Sub. | 76.44% (0.1%) | 77.12% (0.3%) | 76.92% (0.8%) | 79.29% (0.1%) | **78.59% (0.5%)** | 78.28% (0.6%) |
| **GIN** | | | | | | |
| Fine Grid ($L=1$) | | 80.58% (4.0%) | | | 71.73% (0.4%) | |
| Random | 80.19% (5.4%) | 81.25% (11.6%) | 81.83% (1.6%) | **75.01% (1.0%)** | **76.96% (1.1%)** | **76.18% (0.9%)** |
| Topk | **80.96% (8.8%)** | **83.08% (23.9%)** | 80.38% (15.5%) | 72.62% (1.7%) | 71.49% (1.1%) | 71.83% (0.2%) |
| Subgraphs | 80.19% (27.6%) | 82.88% (24.5%) | **82.88% (7.7%)** | 71.71% (1.2%) | 73.03% (0.7%) | 73.01% (1.7%) |
| Rand. & Sub. | 80.58% (14.7%) | 82.12% (3.8%) | 81.35% (3.0%) | 72.64% (0.4%) | 73.71% (0.8%) | 74.16% (1.3%) |
| **GAT** | | | | | | |
| Fine Grid ($L=1$) | | 71.44% (3.9%) | | | 79.09% (1.1%) | |
| Random | **79.04% (3.9%)** | **77.98% (2.1%)** | 72.98% (3.0%) | **79.10% (0.5%)** | **80.28% (0.8%)** | 79.19% (0.4%) |
| Topk | 71.83% (1.2%) | 72.31% (1.5%) | 72.31% (0.1%) | 79.00% (0.4%) | 77.90% (0.9%) | 76.09% (0.9%) |
| Subgraphs | 70.67% (2.0%) | 69.52% (0.6%) | 71.73% (3.0%) | 78.45% (2.3%) | 77.94% (0.6%) | 76.81% (0.8%) |
| Rand. & Sub. | 72.98% (3.9%) | 74.13% (1.9%) | **76.15% (3.1%)** | 78.93% (0.6%) | 79.20% (0.2%) | **79.60% (0.9%)** |

Table 16: Comparison of different coarsening methods on the DBLP and Facebook datasets. The highest (best) performing approach in terms of test accuracy (%) is highlighted in **black**.

| Method ↓ / Dataset → | DBLP | | | Facebook | | |
|---|---|---|---|---|---|---|
| | $L=2$ | $L=3$ | $L=4$ | $L=2$ | $L=3$ | $L=4$ |
| **GCN** | | | | | | |
| Fine Grid ($L=1$) | | 86.48% (0.4%) | | | 74.88% (0.6%) | |
| Random(p=1) | **87.13% (0.1%)** | **87.05% (0.1%)** | **87.08% (0.1%)** | **75.87% (0.4%)** | 75.25% (0.3%) | **75.25% (0.7%)** |
| Random(p=2) | 85.58% (0.0%) | 85.30% (0.3%) | 85.47% (0.5%) | 75.50% (0.4%) | 74.38% (0.0%) | 73.76% (0.3%) |
| Topk(p=1) | 86.79% (0.2%) | 86.82% (0.1%) | 86.03% (0.2%) | 75.25% (0.5%) | 74.75% (0.4%) | 74.75% (0.4%) |
| Topk(p=2) | 85.75% (0.1%) | 85.78% (0.1%) | 85.86% (0.0%) | 74.38% (0.3%) | 74.63% (0.2%) | 74.88% (0.8%) |
| Subgraphs(p=1) | 86.63% (0.1%) | 86.88% (0.3%) | 86.77% (0.0%) | 74.63% (0.6%) | 74.75% (0.3%) | 74.88% (0.3%) |
| Subgraphs(p=2) | 85.86% (0.2%) | 85.75% (0.1%) | 85.78% (0.1%) | 74.88% (0.3%) | 75.25% (0.6%) | 75.12% (0.3%) |
| Rand. & Sub.(p=1) | 86.68% (0.1%) | 86.71% (0.2%) | 86.77% (0.2%) | 74.63% (0.3%) | **75.37% (0.4%)** | 75.00% (0.3%) |
| Rand. & Sub.(p=2) | 85.64% (0.2%) | 85.50% (0.2%) | 85.84% (0.2%) | 74.75% (0.2%) | 74.75% (0.5%) | 75.12% (0.3%) |
| **GIN** | | | | | | |
| Fine Grid ($L=1$) | | 84.82% (0.4%) | | | 73.89% (0.3%) | |
| Random(p=1) | 84.65% (0.3%) | 84.20% (0.2%) | 84.68% (0.3%) | **75.62% (0.3%)** | 74.38% (3.8%) | 74.88% (2.7%) |
| Random(p=2) | **85.27% (0.3%)** | 85.16% (0.3%) | **85.55% (0.3%)** | 74.88% (0.4%) | 75.62% (0.8%) | **76.11% (1.0%)** |
| Topk(p=1) | 84.99% (0.7%) | 84.71% (0.1%) | 84.09% (0.3%) | 74.75% (0.7%) | 74.01% (3.3%) | 73.89% (6.0%) |
| Topk(p=2) | 84.26% (0.1%) | 84.99% (0.4%) | 85.47% (0.1%) | 74.50% (0.7%) | 74.13% (20.3%) | 71.78% (16.9%) |
| Subgraphs(p=1) | 84.85% (0.2%) | 84.00% (0.2%) | 83.80% (0.1%) | 74.88% (0.7%) | 74.75% (0.3%) | 75.25% (1.0%) |
| Subgraphs(p=2) | 83.04% (0.6%) | **85.21% (1.2%)** | 84.88% (1.0%) | 74.13% (0.3%) | 73.14% (0.4%) | 72.77% (1.0%) |
| Rand. & Sub.(p=1) | 84.73% (0.5%) | 84.45% (0.4%) | 83.86% (0.1%) | 75.25% (0.5%) | **75.87% (0.7%)** | 75.62% (1.5%) |
| Rand. & Sub.(p=2) | 84.51% (1.1%) | 85.16% (0.1%) | 84.48% (0.1%) | 74.75% (0.7%) | 74.63% (0.2%) | 70.79% (19.8%) |
| **GAT** | | | | | | |
| Fine Grid ($L=1$) | | 85.16% (0.2%) | | | 72.65% (0.7%) | |
| Random(p=1) | 84.62% (0.2%) | 84.71% (0.3%) | 84.59% (0.3%) | 74.38% (0.4%) | **73.76% (0.7%)** | 73.02% (0.5%) |
| Random(p=2) | 85.58% (0.3%) | 85.05% (0.3%) | 85.21% (0.5%) | 73.64% (0.6%) | 71.91% (0.4%) | 71.53% (0.4%) |
| Topk(p=1) | 85.61% (0.3%) | 85.07% (0.4%) | **85.78% (0.7%)** | 73.27% (1.3%) | 72.15% (0.8%) | 72.15% (1.1%) |
| Topk(p=2) | 85.86% (0.4%) | 85.05% (0.2%) | 84.42% (0.5%) | 72.40% (0.7%) | 73.14% (1.8%) | 72.90% (1.2%) |
| Subgraphs(p=1) | 84.76% (0.1%) | 84.71% (0.2%) | 84.59% (0.4%) | 71.91% (3.1%) | 71.16% (0.8%) | 71.66% (0.7%) |
| Subgraphs(p=2) | 85.84% (0.3%) | 85.16% (0.2%) | 85.30% (0.1%) | 72.15% (2.2%) | 71.16% (1.9%) | 71.66% (2.0%) |
| Rand. & Sub.(p=1) | 85.86% (0.3%) | 85.30% (0.2%) | 85.47% (0.4%) | **74.38% (1.3%)** | 72.65% (0.2%) | **73.51% (0.7%)** |
| Rand. & Sub.(p=2) | **85.86% (0.3%)** | **85.30% (0.2%)** | 85.47% (0.4%) | 72.28% (1.6%) | 72.40% (0.1%) | 70.54% (1.0%) |

Table 18: Experimental setup for transductive learning datasets.

| Component | Details |
|---|---|
| Network architecture | GCN Kipf & Welling (2016) with 4 layers and 192 hidden channels. GIN Xu et al. (2018) with 3 layers and 256 hidden channels. GAT Velickovic et al. (2017) with 3 layers and 64 hidden channels, using 2 heads |
| Loss function | Negative Log Likelihood Loss except for PPI (transductive) and Facebook Yang et al. (2020) which use Cross Entropy Loss |
| Optimizer | Adam Kingma & Ba (2014) |
| Learning rate | $1 \times 10^{-3}$ |
| Baseline training | 2000 epochs |
| Mulstiscale gradients level | 2, 3, and 4 levels |
| Coarse-to-fine strategy | Each coarsening reduces the number of nodes by half compared to the previous level |
| Sub-to-Full strategy | Using ego-networks Gupta et al. (2014). 6 hops on level 2, 4 on level 3, and 2 on level 4 |
| Multiscale training strategy | Using [1000, 2000] epochs for 2 levels, [800, 1600, 3200] epochs for 3 levels, and [600, 1200, 2400, 4800] epochs for 4 levels (the first number is the fine grid epoch number) |
| Evaluation metric | Accuracy |
| Timing computation | GPU performance. An average of 100 epochs after training was stabled |

Table 19: Comparisson with results as showes in Zeng et al. (2020), in Micro F1 score

| | Flickr |
|---|---|
| GCN | 0.492 |
| GraphSAGE | 0.501 |
| FastGCN | 0.504 |
| S-GCN | 0.482 |
| AS-GCN | 0.504 |
| ClusterGCN | 0.481 |
| GraphSAINT-Node | 0.507 |
| GraphSAINT-Edge | 0.510 |
| GraphSAINT-RW | 0.511 |
| GraphSAINT-MRW | 0.510 |
| Ours | **0.538** |

# H Additional Results for Shapenet Dataset

We test out methods on the full Shapenet dataset Chang et al. (2015) using DGCNN Wang et al. (2018) and demonstrate the results of our multiscale training process in Table 20. We observe that our method is not limited to the 'Airplane' category of this dataset, and we achieved comparable results across various connectivity values and pooling methods while using a significantly cheaper training process.

To emphasize the efficiency of our methods, we further provide timing analysis of Shapanet 'Airplanes' data computations. Timing was calculated using GPU computations, and we averaged across 5 epochs after the training loss was stabilized. Results appear in Table 21 for multiscale training and in Table 22 for multiscale gradient computations, where we show that our training process was indeed faster than the original process, which only took into consideration the fine grid. We provide additional experimental details in Table 23.

Table 20: Results on full Shapenet dataset using DGCNN. Results represent Mean test IoU, using **multiscale training**.

| Graph Density (kNN) | Fine Grid | Random | Subgraphs | Rand. & Sub. |
|:---:|:---:|:---:|:---:|:---:|
| k=6 | 77.86% (0.07%) | 79.14% (0.17%) | 77.92% (0.41%) | 77.96% (0.07%) |
| k=10 | 79.70% (0.44%) | 79.78% (0.32%) | 79.41% (0.12%) | 79.42% (0.25%) |
| k=20 | 80.16% (0.30%) | 80.75% (0.18%) | 80.51% (0.23%) | 80.67% (0.22%) |

Table 21: Average epoch time for Shapenet 'Airplane' dataset using DGCNN, using **multiscale training**. Time is calculated in $mili-seconds$.

| Graph Density (kNN) | Fine Grid | Random | Subgraphs | Rand. & Sub. |
|:---:|:---:|:---:|:---:|:---:|
| k=6 | 18,596 | 13,981 | 15,551 | 14,760 |
| k=10 | 20,729 | 15,409 | 17,015 | 16,288 |
| k=20 | 29,979 | 21,081 | 22,646 | 21,891 |

Table 22: Average epoch time for Shapenet 'Airplane' dataset using DGCNN, using **multiscale gradient computation training**. Time is calculated in $mili-seconds$.

| Graph Density (kNN) | Fine Grid | Random | Subgraphs | Rand. & Sub. |
|:---:|:---:|:---:|:---:|:---:|
| k=6 | 44,849 | 28,666 | 30,265 | 29,465 |
| k=10 | 54,064 | 34,528 | 36,094 | 35,299 |
| k=20 | 89,923 | 58,840 | 60,660 | 59,814 |

Table 23: Experimental setup for ShapeNet dataset.

| Component | Details |
| --- | --- |
| Dataset | ShapeNet Chang et al. (2015), 'Airplanes' category |
| Network architecture | DGCNN Wang et al. (2018) with 3 layers and 64 hidden channels |
| Loss function | Negative Log Likelihood Loss |
| Optimizer | Adam Kingma & Ba (2014) |
| Learning rate | $1 \times 10^{-3}$ |
| Batch size | 16 for multiscale training, 12 for multiscale gradient computation |
| Multiscale levels | 3, each time reducing graph size by half |
| Epochs per level | Fine training used 100 epochs. Multiscale training used [40, 80, 160] epochs for the 3 levels. |
| Mulstiscale gradients level | 2, coarse graph reduces graph size by 0.75% |
| Multiscale gradients strategy | 50 epochs using multilevels gradients, 50 epochs using original fine data |
| Evaluation metric | Mean IoU (Intersection over Union) |

## I  Additional Inductive Learning Datasets

The PPI dataset Zitnik & Leskovec (2017), is a biological graph dataset where nodes represent proteins, edges indicate interactions between them, and each protein is associated with multiple labels describing its biological functions. includes 20 graphs for training, 2 for validation, and 2 for testing, with an average of 2,372 nodes per graph. Each node has 50 features and is assigned to one or more of 121 possible labels. We use the preprocessed data from Hamilton et al. (2017) and present our results in Table 24. In this experiment, we use the Graph Convolutional Network with Initial Residual and Identity Mapping (GCNII) Chen et al. (2020). The details of the network architecture and the experiment are available in Table 25. Our results demonstrate that the Multiscale Gradients Computation method achieves comparable accuracy across various pooling techniques while significantly reducing training costs.

We further evaluate our methods on the NCI1 (Wale et al., 2008) and ogbg-molhiv (Hu et al., 2020) datasets, which are inductive learning benchmarks consisting of multiple graphs (4,110 for NCI1 and 41,127 for ogbg-molhiv). NCI1 consists of chemical compounds represented as graphs, where nodes are atoms and edges are chemical bonds, with the goal of predicting whether a compound is active against specific cancer cell lines, while MolHIV is a molecular property prediction dataset, where the task is to predict whether a molecule inhibits HIV replication based on its molecular structure. The task for both datasets is graph classification, and we note that as stated in Coupette et al. (2025) those datasets are considered to be informative datasets. Our results, presented in Table 26 demonstrate that on these multi-graph datasets and for this type of task, our methods achieve performance that is comparable to, and in some cases surpasses, the baseline results.

Table 24: Results on PPI (inductive) dataset using GCNII and multiscale gradient computation method.

| Method | F1 Score |
|---|---|
| Fine Grid | 99.27% (0.11%) |
| Random | 99.18% (0.02%) |
| Topk | 99.23% (0.04%) |
| Subgraph | 99.21% (1.40%) |
| Rand. & Sub. | 99.20% (0.01%) |

Table 25: Experimental setup for PPI dataset.

| Component | Details |
|---|---|
| Dataset | PPI Zitnik & Leskovec (2017) |
| Network architecture | GCNII Chen et al. (2020) with 9 layers and 2048 hidden channels |
| Loss function | Binary Cross Entropy with logits |
| Optimizer | Adam Kingma & Ba (2014) |
| Learning rate | $5 \times 10^{-4}$ |
| Wight initialization | Xavier Glorot & Bengio (2010) |
| Batch size | 1 |
| Baseline training | 1000 epochs |
| Mulstiscale gradients level | 2, coarse graph reduces graph size by 0.75% |
| Multiscale gradients strategy | 500 epochs using multilevels gradients, 500 epochs using original fine data |
| Evaluation metric | F1 |

Table 26: Results on NCI1 and MolHIV datasets using GCN and multiscale training with 2 levels.

| Method | NCI1 (Acc.) | MolHIV (AUC-ROC) |
|---|---|---|
| Fine Grid | 69.37% (0.42%) | 70.39% (0.76%) |
| Random | 70.45% (2.14%) | 70.41% (1.52%) |
| Topk | 69.37% (0.83%) | 70.40% (1.62%) |
| Subgraph | 69.73% (1.70%) | 70.94% (1.59%) |
| Rand. & Sub. | 68.56% (0.88%) | 71.01% (1.35%) |

## J   Ablation Study

We conduct an ablation study on the PubMed dataset Sen et al. (2008). In Table 27, we evaluate the impact of different coarsening ratios $m$ between levels. The results show that performance remains relatively stable across a range of values. For our main experiments, we choose $m = \frac{1}{2}$, as it balances two trade-offs: minimal coarsening may not yield significant training efficiency gains, while excessive coarsening—especially with multiple levels—can degrade performance.

In Table 28, we analyze how the allocation of training epochs across levels affects results. We compare our default strategy, which doubles the number of epochs at each coarsening level, with two alternatives: using a constant number of epochs per level, and adding a fixed number of epochs at each level. All strategies yield similar accuracy, but we adopt the exponential schedule, as it emphasizes learning on coarser levels, where training is more efficient.

Table 27: Various coarsening ratios for using PubMed dataset with GCN and 3 levels of coarsening.

|  | $m = \frac{1}{4}$ | $m = \frac{1}{3}$ | $m = \frac{1}{2}$ | $m = \frac{4}{3}$ | $m = \frac{3}{2}$ |
|---|---|---|---|---|---|
| Random | 77.8% (0.4%) | 77.7% (0.2%) | 77.5% (0.6%) | 77.3% (0.5%) | 77.4% (0.3%) |
| Topk | 76.4% (0.5%) | 77.5% (0.8%) | 78.2% (0.3%) | 76.9% (0.6%) | 77.5% (0.8%) |
| Subgraphs | 78.0% (0.3%) | 77.9% (0.5%) | 78.0% (0.3%) | 78.0% (0.7%) | 77.8% (0.4%) |
| Rand. & Sub. | 78.2% (0.3%) | 78.0% (0.5%) | 77.8% (0.8%) | 78.1% (0.3%) | 77.7% (0.4%) |

Table 28: Various epoch distributions using the PubMed dataset with GCN and 3 levels of coarsening.

|  | Default | Constant | Additive |
|---|---|---|---|
| Random | 77.5% (0.6%) | 77.20% (0.3%) | 78.10% (0.6%) |
| Topk | 78.2% (0.3%) | 76.80% (0.4%) | 76.70% (0.3%) |
| Subgraphs | 78.0% (0.3%) | 78.50% (0.8%) | 77.60% (0.4%) |
| Rand. & Sub. | 77.8% (0.8%) | 77.40% (0.3%) | 78.00% (0.6%) |

