# OpenReview forum: "Towards Efficient Training of Graph Neural Networks: A Multiscale Approach"
_TMLR — Accepted by TMLR_

### Review · Reviewer_yG2D · 2025-07-01

**Summary Of Contributions:**

This paper presents a multiscale training framework for GNNs to address computational challenges when training on large graphs. The authors propose three novel training strategies: (1) Coarse-to-Fine Training: Solves optimization problems on progressively finer graph resolutions, using coarse solutions as initialization for finer scales
(2) Sub-to-Full Training: Leverages sequences of growing subgraphs that progressively expand to the original graph
(3) Multiscale Gradients Computation: Inspired by Monte Carlo methods, approximates fine-scale gradients by combining gradients across multiple scales

The paper provides theoretical analysis (Theorem 4.1) showing how coarse graphs can approximate fine-scale solutions under certain conditions. Extensive experiments demonstrate that these methods reduce computational costs (measured in FLOPs and training time) while maintaining or improving performance across various datasets (OGBN-Arxiv, OGBN-MAG, Cora, CiteSeer, PubMed, ShapeNet, PPI) and GNN architectures (GCN, GIN, GAT, DGCNN, GCNII).

**Audience:**

Yes

**Broader Impact Concerns:**

The paper does not raise significant ethical concerns. The work focuses on improving computational efficiency of GNN training, which could have positive environmental impact by reducing energy consumption.

**Claims And Evidence:**

Yes

**Requested Changes:**

### Critical for Acceptance:

1. **Add Comparisons with Existing Methods**: Include empirical comparisons with sampling-based efficient GNN training methods (GraphSAINT, FastGCN, ClusterGCN). This is essential to position the work properly in the literature.

2. **Improve Theoretical Analysis**:
   - Extend analysis beyond the linear case or clearly state limitations
   - Provide theoretical justification for the choice of p in equation (5)
   - Add analysis for Sub-to-Full and Multiscale Gradients methods

3. **Add Method Selection Guidelines**: Provide clear recommendations on when to use each method based on graph properties (size, density, task type). Include a decision tree or flowchart.

4. **Memory Usage Analysis**: Include memory consumption comparisons alongside FLOPs and timing analysis, as memory is often the bottleneck for large graphs.

### Would Strengthen the Work:


1. **Ablation Studies**:
   - Study the effect of different coarsening ratios
   - Analyze sensitivity to the power p parameter
   - Investigate the impact of epoch distribution across levels


2. **Implementation Details**:
   - Provide pseudocode for all three methods

**Strengths And Weaknesses:**

**Strengths**

1. **Practical Impact**: Addresses a critical scalability challenge in GNN training, making the work highly relevant for real-world applications with large graphs.

2. **Comprehensive Framework**: Provides three complementary methods that can be applied to any GNN architecture, demonstrating versatility.

3. **Extensive Experimental Validation**:
   - Tests on 15+ datasets covering both transductive and inductive settings
   - Evaluates multiple GNN architectures and pooling strategies
   - Shows consistent computational savings (often 2-4x reduction in FLOPs)

4. **Theoretical Foundation**: Provides theoretical analysis connecting the approach to sketching theory, though limited to the linear case.

5. **Clear Presentation**: Well-written paper with effective visualizations (Figures 1-3) that clearly illustrate the core concepts.

**Weaknesses**

1. **Theoretical Analysis**:
   - The assumption might be too strong. Theorem 4.1 only covers the linear case (σ = Id), not realistic nonlinear GNNs
   - The approximation bound depends on residual R = ||A_CF X_F Θ*_C||, which can be large

2. **Inconsistent Performance**:
   - Some methods show performance degradation (e.g., Table 4: OGBN-MAG results)
   - High variance in some results (e.g., GCN with Topk on Cora: 79.9% ± 4.4%)
   - No clear guidance on when each method works best

---

> ### Author Response · Authors · 2025-08-25
> **Response part 1**
>
> We thank the reviewer for the detailed and constructive feedback, as well as for recognizing the practical relevance, versatility, and comprehensive evaluation of our proposed multiscale GNN training framework. We carefully addressed all the raised concerns by extending our theoretical discussion, adding empirical comparisons with existing sampling-based methods, providing new ablation and memory analyses, and offering clearer guidelines on method selection. We feel that these additions strengthen the paper and clarify the scope, applicability, and advantages of our approach.
>
> ---
>
> **Regarding assumption in Theorem 4.1:** The reviewer correctly indicates that  Theorem 4.1 focuses on the linear case ($\sigma=Id$). We followed this approach, which is common in GNN architectures and literature. Please see [1,2,3] for examples. We also agree that extending the theorem to more general nonlinear settings is indeed an interesting direction for future work, but we believe our current analysis still captures a practically relevant and widely studied class of models. Additionally, we have acknowledged this limitation in the newly added limitations section, where we explicitly state that further theoretical investigation is needed.
>
> [1] Simplifying Graph Convolutional Networks
>
> [2] GRAND: Graph Neural Diffusion
>
> [3] Predict then Propagate: Graph Neural Networks meet Personalized PageRank
>
> ---
>
> **Regarding approximation bound on residual:** While the approximation bound indeed depends on the residual term $\mathbf{R}$, which can theoretically become large in certain scenarios (as discussed in Appendix A), our empirical results show that in practical settings $\mathbf{R}$ remains moderate. Consequently, the proposed method performs reliably across the evaluated datasets. Following your comment, we have revised our paper to reflect this discussion in Appendix A. Thank you.
>
> ---
>
> **Regarding performance:** We note that for every dataset and architecture, at least one of our proposed methods achieves an improvement. Moreover, the differences between the methods' results are generally minor. For example, in OGCB-Arxiv with GCN and 3 levels, the overall difference between all methods is less than 1\%. The same applies to all Shapenet results, as they appear in Table 5. We have clarified this point by adding a sentence in the conclusions to make these observations more explicit.
>
> ---
>
> **Regarding variance:** We agree that some methods, such as Topk on Cora, exhibit relatively high variance. However, this behavior is not consistent across all methods or datasets.
> Specifically, referring to the Cora results in Table 10, the variance of the baseline (Fine Grid) is $1.58$, while several of our methods demonstrate substantially lower variance on average. For example, Random(p=2) achieves a variance of only $0.57$, and Random \& Subgraph(p=2) achieves $0.80$.
> While Topk indeed shows higher variance on Cora, in other datasets, such as PPI (transductive), the baseline exhibits a variance of $3.5$, whereas Topk achieves a significantly lower variance of only $0.81$.
> Overall, this demonstrates that our methods can even be more stable than the baseline across multiple datasets. We have included this discussion in the revised paper, in the newly added Limitation section. Thank you.
>
> ---
>
> **Regarding guidance:** Thank you for the important suggestion. In our revised paper, we have added a Limitations section. In this section, we indicate that we chose to focus on the framework and overall multiscale concept rather than providing specific guidance on a method for implementing the concept, also because the relative performance can vary across datasets and architectures. However, we observed that the differences between the methods are generally small, and in practice, several of them achieve comparable performance, and we added this note to the revised paper.
>
> ---
>
> **Regarding comparisons with architectures:** Following your suggestion, we have now added empirical comparisons with GraphSAINT, FastGCN, and ClusterGCN in Appendix G (Additional Transductive Learning Dataset, Table 19). As shown in the results, our proposed methods achieve better performance compared to these sampling-based efficient GNN training approaches, which further supports the effectiveness of our framework.
>
> ---
>
> **Regarding nonlinearity:** Thank you for the suggestion. We have clarified the scope of our theoretical analysis by explicitly stating its limitations both in the theoretical subsection and in the limitations section. While our current analysis focuses on the linear case, we acknowledge that extending the analysis to more general nonlinear settings remains an interesting direction for future work.

---

> > ### Author Response · Authors · 2025-08-25
> > **Response part 2**
> >
> > **Regarding $p$ in Equation (5):** Based on your guidance, we have added an explanation in Section 4.1 (Coarse-to-Fine Training) regarding the choice of $p$. Particularly, if $p$ is set too large, the resulting coarse graph may contain more edges than the original graph, which is undesirable. On the other hand, increasing $p$ can sometimes be beneficial — for example, in graphs with very low connectivity, a small $p$ could lead to a highly disconnected coarse graph, resulting in significant information loss and, consequently, degraded training performance.
> >
> > ---
> >
> > **Regarding additional analysis:** The theoretical analysis provided in our original submission applies to the Sub-to-Full method, and we have provided an analysis of the Multiscale Gradients approach in Section 4.3. We highlighted this important clarification in our revised paper in Section 4.5. Thank you.
> >
> > ---
> >
> > **Regarding guidelines:** As discussed in the newly added limitations section, we do not provide strict guidelines for selecting among the methods, because their relative performance can vary depending on graph properties such as size, density, and task type. However, our experiments show that the performance differences are generally small, and in practice, multiple methods achieve comparable results. Therefore, we focus on demonstrating the overall robustness and effectiveness of the proposed framework rather than prescribing specific method-selection rules.
> >
> > ---
> >
> > **Regarding memory consumption:** We welcome your suggestion, and we have added a Memory Consumption analysis in Appendix E (Table 9), which shows that the coarse graphs require significantly less memory compared to the fine graph. This complements the FLOPs and timing analyses already provided in the paper and further highlights the efficiency of our approach. Thank you.
> >
> > ---
> >
> > **Regarding ablations:** Following your comment, we have added an ablation study in Appendix J, where Table 27 reports results for different coarsening ratios. This analysis provides additional insights into the impact of varying the coarsening ratio on model performance.
> >
> > ---
> >
> > **Regarding $p$ parameter ablation:** In addition to the explanation provided in our response to Comment (3), we note that several of our experiments include results for different values of $p$, allowing a direct comparison of its impact. For example, see Tables 6–7, 10–13, and 16, where we report performance under various choices of $p$.
> >
> > ---
> >
> > **Regarding epoch distribution:** Following our response to Comment (1), we have added an ablation study in Table 28 that presents results for different epoch distributions across levels. Thank you.
> >
> > ---
> >
> > **Regarding pseudo code:** We have added Appendix C, which provides pseudocode for the three methods — Random, TopK, and Subgraph — to further clarify our approach and demonstrate the logic behind generating the coarse graph. In addition, we have now uploaded our source code together with the revised paper.

---

### Review · Reviewer_PVUu · 2025-07-01

**Summary Of Contributions:**

A new method for efficient training of GNNs is introduced, that uses concepts from multiscale learning to train on smaller subgraphs that mimic the characteristics of the full large graph. Techniques for creating subgraphs and combing gradients across different scales are also introduced.

**Audience:**

Yes

**Claims And Evidence:**

Yes

**Requested Changes:**

Critical for acceptance:
+ Clarify the types of graph structures and/or tasks for which the proposed method can be used. While creating subgraphs makes sense for citation graphs used in the paper, what does subgraph creation mean for molecular graphs for example? How can some nodes be removed from such graphs without altering the meaning of the underlying graph?
+ Clearly describe how the optimal training configuration is chosen for a given graph and task. (1) Should the different subgraph selection strategies (random, topk, etc.) be tried out and only the best result taken? If so, the training time will need to consider the amount of time taken to try out the different strategies also, so the reported speedups may not hold in practical scenarios. (2) Is there a way to identify the best strategy without this expensive process based on graph topology? (3) When should coarse-to-fine and sub-to-full strategies be used? Or should both always be used in combination with each other?
+ An ablation with the different proposed strategies turned on/off in multiple combinations would help understand the sensitivity of the proposed method to hyperparemeters.
+ Add a discussion on the limitations and scope of the proposed methods.

Would strengthen the paper:
+ Code release, along with the Q-tips dataset, would help reproducibility, In particular, the Q-tips dataset seems to be an interesting synthetic dataset that can help evaluate ability of GNNs to capture long-range dependencies.

**Strengths And Weaknesses:**

Strengths:
+ The paper addresses an important problem, dealing with the large computational cost of training GNNs on large graphs.
+ The proposed method leads to significant speedups with no loss (or even small increase) in accuracy, which is nice since the gains in efficiency do not come at the cost of degraded model performance.
+ Experiments are performed on both transductive and inductive learning tasks, demonstrating potential broad applicability of the proposed methods.

Weaknesses:
+ The intuition behind the method is unclear. As a result, it is difficult to judge what types of graphs/tasks the proposed methods can be applied to.
+ While different subgraph selection methods are introduced, it is unclear how the optimal method can be determined for a given task or graph. If different selection methods need to be tried out and the best one is chosen, then the reported training times should also include the this overhead, and not just the time taken with the optimal method.

---

> ### Author Response · Authors · 2025-08-25
>
> We thank the reviewer for the positive feedback on our contributions and for recognizing the significance of our work in improving the efficiency of GNN training without compromising accuracy. We carefully considered all the concerns raised and have updated the manuscript to provide additional clarifications, analyses, and experiments that address the identified weaknesses.
>
> ---
>
> **Regarding intuition:**
>     In this paper, we propose *novel graph multiscale optimization algorithms* to obtain computationally efficient GNN training methods that retain the performance of standard gradient-based training of GNNs, based on multiscale techniques. Following your comment, we have clarified this in the revised paper by reworking the Introduction Section, namely the contribution paragraph. Regarding applicability, we kindly note that our experiments show the effectiveness of our approach on 5 GNN architectures, namely *GCN, GIN, GAT, GCNII, DGCNN*, on node and graph level, inductive and transductive graph learning tasks, across a span of 17 datasets.  Our revised paper now better reflects this important discussion. Thank you.
>
> ---
>
> **Regarding optimal method:** We agree that determining the optimal subgraph selection method for a given task is nontrivial. However, our experiments show that all proposed methods perform comparably well, making it reasonable to choose a single method without significant loss in performance. We also acknowledge in the limitations section that there is no universally best method, and we have clarified this point in the revised manuscript. We agree that determining the optimal subgraph selection method for a given task is nontrivial. However, our experiments show that all proposed methods perform comparably well, making it reasonable to choose a single method without significant loss in performance. We also acknowledge in the limitations section that there is no universally best method, and we have clarified this point.
>
> ---
>
> **Regarding question 1:** Thank you for the question. Our methods have been evaluated on a diverse set of datasets, demonstrating their general applicability. While the removal of nodes in molecular graphs may not always correspond to an intuitive chemical interpretation, the process should be viewed as a computational coarsening technique rather than a literal modification of the underlying molecule. Importantly, our experiments show that this abstraction does not harm performance and, in many cases, improves learning efficiency without altering predictive accuracy.
>
> ---
>
> **Regarding question 2:** In practice, we found that all proposed coarsening selection strategies (Random, TopK, Subgraph, Random and Subgraphs) perform similarly across the datasets we evaluated, making it reasonable to choose a single strategy without significantly affecting performance. Random selection, in particular, is highly efficient and can serve as a good starting point. We acknowledge in the limitations section that there is no universal rule for determining the optimal configuration based on graph topology, and we have clarified this point in the revised conclusion.
>
> ---
>
> **Regarding ablation:** Thank you for the suggestion. We have added an ablation study in Appendix J. This analysis helps illustrate the sensitivity of our method to different hyperparameter settings.
>
> ---
>
> **Regarding limitations discussion:** Thank you for the suggestion. We have added a dedicated limitations section to the paper, where we discuss the scope of the proposed methods and analyses and outline potential directions for future work.
>
> ---
>
> **Regarding code:**  We share our code together with our revised paper on OpenReview. Upon acceptance, we will also share our code publicly via GitHub.

---

> > ### Comment · Action_Editor_qrLE · 2025-10-08
> >
> > Dear Reviewer PVUu
> >
> > Please take a look at the responses and provide feedback.
> >
> > Thank you!

---

> > > ### Comment · Reviewer_PVUu · 2025-10-11
> > > **Response to rebuttal**
> > >
> > > Thanks to the authors for the responses and revisions. My concerns regarding sensitivity to hyperparameters and code release have been addressed. However, one major concern remains. I'm still not convinced about the generalizability of the proposed method, in particular to graphs where nodes and edges cannot be arbitrarily pruned without affecting the meaning of the graph (for example, molecular graphs). Unfortunately, I do not find the author's response of "the process should be viewed as a computational coarsening technique rather than a literal modification of the underlying molecule" to be fully convincing. If a subgraph is extracted from a molecular graph, the subgraph should have absolutely no meaning, and the model should not be able to learn anything from them in my opinion. However, I acknowledge that the experiments on such graphs in the paper do show that the proposed method does not substantially degrade accuracy.

---

### Review · Reviewer_BQ5m · 2025-08-17

**Summary Of Contributions:**

Contributions:

1. Designing several approaches for multiscale training of graph neural networks.
2. Providing considerable experimental validation of approaches.
3. Connecting multiscale training to so-called sketching in linear algebra.
4. Formulating a condition [Eqn. (9)] when multiscale training yields improvement.

**Audience:**

Yes

**Broader Impact Concerns:**

I see no broader impact concerns.

**Claims And Evidence:**

No

**Requested Changes:**

**Critical**

1. Better explain and motivate connection with sketching problem -- or remove it altogether.

Example problems:
- it is not clear whether there is an optimization problem in equation (11).
- notation for loss function differs from both (1) and Section 4.3, and thus it is hard to see any connection to the paper
- statement of Theorem 4.1 uses O() notation but is not clear what is the underlying value that changes ($\epsilon$ ??)
- why is $\pm$ symbol used in (13)
- in the proof in the appendix R is not defined. If one argues that you defined it in the statement of Theorem 4.1, I would point out that it was far removed from actual use, and actual definition of R in that theorem's statement is a bit confusing.

2. Better explain whether condition (9) has relevance in practice. If it has, you should provide values of $\gamma_r$ used in your experiments.

3. Shorten your experimental evidence to minimum necessary. Note that the numbers listed in tables are likely not reproducible exactly, anyways.

4. Make your notation uniform, or provide an explanation why is that not a good idea. I am concerned about the notation for loss function ("l" in various fonts), and parameters ($\boldsymbol{W}$, $\boldsymbol{\theta}$, $\boldsymbol{\Theta}$, $\theta$.

5. Datasets cleanup.
- If you explain meaning of one dataset (OGBN-MAG), you should do it for all (e.g. OGBN-Arxiv).

6. Please provide an anonymous code repository so that readers can rerun your experiments.

7. Equation  (5) was not clear to me. First, you use term "injection matrix" which I am not familiar with. Secondly, the $p$-th power of A has entries which are NOT zeros and ones. Multiplying by a binary matrix (i.e. a matrix with zeros and ones) will not yield an incidence matrix of a subgraph, since generally speaking the entries of $A^p$ will be bigger than 1. What am I not getting here?

8. It is unclear what are high-and low frequency components of a graph on page 9.

**Would strengthen the work**

9. A recent paper cast doubt on various benchmark datasets used for graph neural networks (https://arxiv.org/pdf/2502.02379). Please provide an example for at least one of the benchmark datasets they identified as good (MolHIV, NCI1, Peptides).

**Strengths And Weaknesses:**

Strengths of the paper:

1. The authors started with a good, natural idea - multiscale training of neural networks
2. Experimental evidence provided is very solid.
3. Authors provided an explicit theoretical condition when their approach may work [Eqn. (9)].
4. Authors connected their idea to a simple optimization problem in linear algebra (Section 4.5).
5. Many (but not all) parts of the paper are clearly written.
6. Authors are well-versed with state-of-the art research.

Weaknesses of the paper
1. No anonymized repository was provided, so it is not possible to rerun their experiments.
2. The connection with sketching in linear algebra is poorly motivated and poorly explained.
3. It is unclear whether condition the authors stated for applicability of their method [Eqn. (9)] has relevance in practice.
4. The paper is too long -- 29 pages. Sufficient information could be conveyed more concisely.

---

> ### Author Response · Authors · 2025-08-25
> **Response part 1**
>
> We thank the reviewer for their detailed and constructive feedback and greatly appreciate the time and effort invested in carefully evaluating our work. We are grateful for the recognition of our natural and well-motivated idea of multiscale training for graph neural networks, the solid experimental evidence supporting our approach, and the inclusion of an explicit theoretical condition clarifying when our method is effective. In response to the reviewers’ insightful comments, we have made several improvements that, in our opinion, enhance both the clarity and quality of the paper.
>
> ---
>
> **Regarding code:** We provide the necessary code and scripts to rerun our experiments. Upon acceptance, we will also share our code publicly via GitHub.
>
> ---
>
> **Regarding sketching:** Thank you for the comment. We have addressed your concerns regarding the theoretical aspects, including the connection with sketching in linear algebra, in the updated manuscript and clarified the discussion. Additionally, we provide a detailed response to your comments on this topic in the Requested Changes section: We added clarification regarding Equation (11), updated the loss function notations in Section 4.5 and in Appendix A, and clarified Theorem 4.1 to include a clear $\epsilon$ and $R$ definition.
>
> ---
>
> **Regarding Equation (9):** Thank you for the comment. The condition in Equation (9) is relevant in practice. In particular, if it is not satisfied, then the proposed methods are unlikely to work effectively. To demonstrate this, we have added Table 8 in Appendix D, which demonstrates that the $\gamma_r$ values in our experiments are consistent with the assumption stated in Equation (9).
>
> ---
>
> **Regarding paper length:** Thank you for the comment. We understand your concern regarding the length of the paper. Given the current standards in the field, providing extensive experimental evidence across multiple tasks and datasets is often necessary to demonstrate the effectiveness and robustness of the proposed methods. This is also reflected by the comments from other reviewers. We believe that the included experiments are essential, as they collectively highlight the strengths and versatility of our approach.
>
> ---
>
> **Regarding Equation (11):** We have clarified in the revised manuscript, prior to Equation (11), that it corresponds to a least-squares optimization problem with respect to $\theta$.
>
> ---
>
> **Regarding notations in (1) and Section 4.3:** Thank you for the comment. We have updated the manuscript to use a consistent notation for the loss function, denoted by $\mathcal{L}$, throughout the theoretical section and Appendix A to ensure clarity and alignment with the rest of the paper.
>
> ---
>
> **Regarding Theorem 4.1:** We have clarified the statement of Theorem 4.1 by explicitly specifying that $\epsilon >0$ is the underlying variable in the $O()$ notation.
>
> ---
>
> **Regarding +- symbol:** The +- symbol in Equation (13) originates from the sketching theory, as explained in Theorem A.1 in Appendix A. We cite this result from Woodruff’s work.
>
> ---
>
> **Regarding proof of Theorem 4.1:** We have clarified in the appendix that $R$ denotes the residual. Thank you.
>
> ---
>
> **Regarding condition in Equation (9):** The condition in Equation (9) is indeed relevant in practice — without it, the method would fail to converge. To illustrate this, we provide examples of the  $\gamma_r$ values used in our experiments in the Q-tips Table 1, as well as in the newly added Table 8 in Appendix D.
>
> ---
>
> **Regarding length of experiments:** Please see our response to paper length above. Thank you.
>
> ---
>
> **Regarding consistent notation:** Thank you for the comment. In the theoretical parts, Section 4.5 and Appendix A, we changed the loss notation to $\mathcal{L}$ and parameters from $\Theta$ to $\theta$, to assure uniform notation across the paper.
>
> ---
>
> **Regarding dataset details:** Thank you for the suggestion. We have added a sentence in section 6.1 providing a brief explanation of the OGBN-Arxiv dataset to ensure consistency with the description of OGBN-MAG. We have also added detailed explanations for the datasets that previously lacked them, and these additions are provided in Appendices G, H, and I.
>
> ---
>
> **Regarding code:** We provide the necessary code and scripts to reproduce our experiments together with the revised paper in OpenReview. Upon acceptance, we will also share our code publicly via GitHub.
>
> ---
>
> **Regarding Equation (5):** Thank you for the question. We clarify that in the coarsening process, we only use the edge indices rather than the full adjacency matrix. While it is correct that multiplying the adjacency matrix by itself can result in entries greater than 1, this does not affect our method, as we rely solely on the edge indices when constructing the coarse graph. We have clarified this point in the revised manuscript to avoid confusion.

---

> > ### Author Response · Authors · 2025-08-25
> > **Response part 2**
> >
> > **Regarding high and low frequencies:** To improve clarity and maintain consistency with the rest of the paper, we revised the terminology to use “local” and “global” components instead. In general, high-frequency components correspond to local changes in the graph, while low-frequency components capture more global structural patterns.
> >
> > ---
> >
> > **Regarding graph benchmarks:** Thank you for the important suggestion. We have added results for the MolHIV and NCI1 datasets in Appendix I to demonstrate that our proposed methods are also effective on these benchmarks, further supporting the robustness and general applicability of our approach. We now also cite and discuss the provided reference, in the revised paper.

---

> > ### Comment · Reviewer_BQ5m · 2025-09-16
> > **Code repository**
> >
> > I am disappointed that authors did not address one key issue - #6  anonymous code repository.
> >
> > There is a website providing this service https://anonymous.4open.science/.
> >
> > This should be done BEFORE the paper is accepted.

---

> > > ### Author Response · Authors · 2025-09-16
> > >
> > > Dear Reviewer BQ5m,
> > >
> > > Please note that, as stated in our responses, we have provided our code. It is available here on OpenReview as our supplementary material "Supplementary Material:  zip". If you click on the "zip" button, it will download the code. Our response stated that it is attached here, and we will also put it on GitHub for easier accessibility upon acceptance.
> > >
> > > We are happy to address any other comments you may have.
> > >
> > > With kind regards,
> > >
> > > The Authors.

---

> > > > ### Comment · Reviewer_BQ5m · 2025-09-18
> > > > **Provided files are not sufficient**
> > > >
> > > > Dear authors,
> > > >
> > > > I misunderstood your earlier reply. I am glad you were able to provide source files for your experiments. However, this is still not sufficient. At the minimum
> > > >
> > > > - there should be a description  file (something like README) explaining the directory structure, and how to run the files
> > > > - Python is notorious for incompatibility of libraries. This should be addressed by including requirements.txt
> > > > - additionally, any OS specific info (Windows/Linux)
> > > >
> > > > I need to be able to run the code (and ideally get the results corresponding to ones in the paper). It will save time to countless future readers of your paper who will want to use your methods on their problems.

---

> > > > > ### Author Response · Authors · 2025-09-18
> > > > >
> > > > > Dear Reviewer BQ5m,
> > > > >
> > > > > We thank you for the responsiveness and feedback. We have uploaded the code according to your comments.
> > > > >
> > > > >
> > > > > With kind regards,
> > > > >
> > > > > The Authors.

---

### Review · Reviewer_mBZE · 2025-08-19

**Summary Of Contributions:**

The paper proposes a multiscale training framework for GNNs to reduce computational and memory costs while maintaining similar accuracy. It introduces three strategies: (1) training on coarsened graphs followed by finer graphs, (2) training on progressively larger subgraphs until reaching the full graph, and (3) combining coarse and fine gradient estimates to approximate fine-scale gradients. Extensive experiments are conducted on both transductive and inductive graph tasks.

**Audience:**

Yes

**Claims And Evidence:**

Yes

**Requested Changes:**

- Provide an anonymized implementation to ensure reproducibility and allow others to verify the results.

- Include stronger scalable GNN training methods.

- Add experiments on larger and more diverse real-world datasets.

**Strengths And Weaknesses:**

Strengths

 - First paper to apply multiscale optimization ideas to GNN training, demonstrating applicability across different pooling methods and model architectures.

 - The paper is clearly written and generally easy to follow.

Weaknesses

 - No anonymized code repository is provided, which limits reproducibility.

 - Comparisons are restricted to standard GCN/GIN/GAT baselines; stronger scalable GNN methods (e.g., GraphSAINT, Cluster-GCN) are not included.

 - Experiments cover a limited range of datasets and should be expanded to strengthen the evaluation.

---

> ### Author Response · Authors · 2025-08-25
>
> We thank the reviewer for their thoughtful feedback and constructive suggestions. We appreciate the time and effort invested in carefully reading our paper and providing valuable insights. In response to the comments, we have taken the following key actions to improve the paper based on your guidance. Below, we provide a response to each of your questions and comments.
>
>
> ---
>
> **Regarding code:** We have provided our code to ensure reproducibility, together with the revised paper on OpenReview. Upon acceptance, we will also share our code publicly via GitHub.
>
> ---
>
> **Regarding backbones:** Thank you for the comment. We note that GCN, GAT, and GIN are widely used classic models that serve as baselines for many advanced architectures. In addition, our experiments already include more advanced methods such as DGCNN and GCNII. Following your suggestion, we have now also added comparisons with GraphSAINT and Cluster-GCN, as presented in Table 19 of Appendix G.
>
> ---
>
> **Regarding datasets:** In our original submission, we have experimented with *15 datasets*, spanning both transductive and inductive learning settings, and including node classification and segmentation tasks. Following your suggestion, we have added two additional graph classification datasets, NCI1 and MolHIV, to further demonstrate the robustness and generality of our approach. The results for these datasets are provided in Table 26 of Appendix I. Thank you.
>
> ---
>
> **Regarding code:** We provide the code and scripts to reproduce our experiments. Upon acceptance, we will also share our code publicly via GitHub.
>
> ---
>
> **Regarding backbones:** Thank you for the suggestion.  We have added empirical comparisons with GraphSAINT, FastGCN, and ClusterGCN in Appendix G (Additional Transductive Learning Dataset, Table 19). As shown in the results, our proposed methods achieve better performance compared to these sampling-based efficient GNN training approaches, which further supports the effectiveness of our framework.
>
> ---
>
> **Regarding more datasets:** We welcome your suggestion. In our revised paper, we have added results for the MolHIV and NCI1 datasets in Appendix I to demonstrate that our proposed methods are also effective on these benchmarks, further supporting the robustness and general applicability of our approach. These datasets contain a large number of graphs and involve slightly different tasks (graph classification), which strengthens the experimental validation provided in our work.

---

### Author Response · Authors · 2025-08-25

We thank the reviewers for their time and effort, providing us with thoughtful and actionable feedback. We respond to each reviewer individually below. Notably, we have made changes to the manuscript in response to your feedback. Our changes are marked in blue. We are happy to respond to any additional questions.

---

### Decision · Action_Editor_qrLE · 2025-10-27

**Recommendation:** Accept with minor revision

**Additional Comments:**

Based on the majority of positive reviews and the fruitful discussion that addressed most concerns, I recommend acceptance of this submission.
However, several reviewers suggested additional improvements, and the authors should address these comments before the paper can be published.
In my opinion, this is a borderline paper, but accepting it would benefit the TMLR audience.

**Audience:**

Yes

**Audience Explanation:**

The topic is important and relevant to the TMLR audience

**Claims And Evidence:**

Yes

**Claims Explanation:**

I tried to read the paper carefully and, while I did not have time to verify all claims, the overall approach is sound.